# How Learnable Grids Recover Fine Detail in Low Dimensions: A Neural Tangent Kernel Analysis of Multigrid Parametric Encodings

**Samuel Audia, Soheil Feizi, Matthias Zwicker & Dinesh Manocha**
University of Maryland, College Park
{sjaudia,sfeizi,zwicker,dmanocha}@umd.edu

## Abstract

Neural networks that map between low dimensional spaces are ubiquitous in computer graphics and scientific computing; however, in their naive implementation, they are unable to learn high frequency information. We present a comprehensive analysis comparing the two most common techniques for mitigating this spectral bias: Fourier feature encodings (FFE) and multigrid parametric encodings (MPE). FFEs are seen as the standard for low dimensional mappings, but MPEs often outperform them and learn representations with higher resolution and finer detail. FFE's roots in the Fourier transform, make it susceptible to aliasing if pushed too far, while MPEs, which use a learned grid structure, have no such limitation. To understand the difference in performance, we use the neural tangent kernel (NTK) to evaluate these encodings through the lens of an analogous kernel regression. By finding a lower bound on the smallest eigenvalue of the NTK, we prove that MPEs improve a network's performance through the structure of their grid and not their learnable embedding. This mechanism is fundamentally different from FFEs, which rely solely on their embedding space to improve performance. Results are empirically validated on a 2D image regression task using images taken from 100 synonym sets of ImageNet and 3D implicit surface regression on objects from the Stanford graphics dataset. Using peak signal-to-noise ratio (PSNR) and multiscale structural similarity (MS-SSIM) to evaluate how well fine details are learned, we show that the MPE increases the minimum eigenvalue by 8 orders of magnitude over the baseline and 2 orders of magnitude over the FFE. The increase in spectrum corresponds to a 15 dB (PSNR) / 0.65 (MS-SSIM) increase over baseline and a 12 dB (PSNR) / 0.33 (MS-SSIM) increase over the FFE.

## 1 Introduction

Recent advancements in computer graphics, scientific machine learning, and the broader class of implicit neural representations (Essakine et al., 2024) rely on the simple coordinate-based multilayer perception (MLP) network, which maps between low dimensional spaces (typically $\mathbb{R}^1$, $\mathbb{R}^2$, or $\mathbb{R}^3$). Despite their diminutive size, coordinate-based MLPs have empowered inverse rendering (Barron et al., 2021; Mildenhall et al., 2020; Müller et al., 2022; Pumarola et al., 2020; Yu et al., 2020; Zhang et al., 2020), implicit surface regression (Wang et al., 2021a), solving the rendering equation (Hadadan et al., 2021), and enabled Physics-Informed Neural Networks (Raissi et al., 2017) in the growing field of scientific machine learning. These networks need to have a small memory and computational footprint to maintain the tight timing required in computer graphics (Müller et al., 2021) and to allow for computing higher order derivatives in scientific machine learning. Though simple MLP networks meet these requirements, in their naive implementation, coordinate-based MLPs suffer from what is known as the spectral bias problem (Basri et al., 2020; 2019; Wang et al., 2020), in which they learn higher frequency details orders of magnitude slower than low frequency details. To understand the cause of this bias, machine learning practitioners turn to the neural tangent kernel (NTK).

Introduced by Jacot et al. (2018), the neural tangent kernel describes the training dynamics of a neural network in its infinite width limit. Though using the NTK directly has been shown to produce

worse results than its finite width counterpart (Li et al., 2020), empirical evaluation of the kernel for a given data set and architecture allows practitioners to apply classical techniques, such as eigenvector decomposition, to better understand the underlying network. In fact, the spectrum of the NTK can be used to show a network's ability to learn high frequency information in finite width networks (Tancik et al., 2020; Wang et al., 2020; Yang & Salman, 2019). The eigenvectors for these finer details correspond to the lowest eigenvalues (Basri et al., 2019), so if network adaptations are able to raise the spectrum as a whole, the network is better able to represent the underlying function. For coordinate based MLPs, this adaption is frequently an encoding that maps the input into a higher dimensional latent space before being passed to the network.

The two most popular encodings are parametric encodings (Hadadan et al., 2021; Müller et al., 2022) and Fourier feature encodings (FFE) (Mildenhall et al., 2020; Tancik et al., 2020; Vaswani et al., 2017; Wang et al., 2020). Parametric encodings use auxiliary data structures with learnable parameters to build a higher dimensional embedding space. These data structures frequently take the form of a grid, with samples being interpolated from fixed grid points and concatenated. Fourier feature encodings, in contrast, contain no learnable features and instead use a series of sines and cosines, similar to that of a Fourier transform, to embed the input into a high dimensional unit hypersphere. Its ease of implementation and amenability to analysis has made the Fourier feature encoding the dominant choice in scientific machine learning (Cuomo et al., 2022; Lu et al., 2019; Raissi et al., 2017); however, in computer graphics, parametric encodings have shown orders of magnitude better performance than Fourier feature encodings (Hadadan et al., 2021; Müller et al., 2022) at the cost of a larger memory footprint. Both encodings have been extensively evaluated, ignoring their NTK spectrum, on graphics problems such as 2D image regression, 3D shape regression, inverse rendering, and radiosity calculations. See Tancik et al. (2020) for the FFE examples and Müller et al. (2022); Hadadan et al. (2021) for MPE examples.

The FFEs popularity has led to multiple investigations of its effect on the NTK spectrum (Tancik et al., 2020; Wang et al., 2020). No such analysis has been known previously for parametric encodings. We remedy this fact, by providing an in depth analysis of the NTK for parametric encodings. Specifically, we make the following contributions:

- We derive the neural tangent kernel for the MPE. Through this derivation, we prove that the multigrid encoding raises the eigenvalue spectrum of the neural tangent kernel as compared to the baseline coordinate based MLP by forming a lower bound on its eigenvalues. Our results provide the first theoretical justification for why MPEs are able to learn finer detail and discontinuities better than networks with no encoding.

- We isolate the superior performance of MPEs to their learnable grid and not their embedding space by evaluating the NTK with and without the contributions from the grid. This is fundamentally different from FFEs, which rely solely on the embedding.

- We empirically evaluate the NTK spectrum for the multigrid parametric encoding, Fourier feature encoding, and baseline identity encoding for 2D image regression on 100 synonym sets in ImageNet (Deng et al., 2009) and 3D implicit surface regression Mescheder et al. (2018) on three meshes from the Stanford graphics dataset Turk & Levoy (1994) to validate our proof. Peak signal-to-noise ratio (PSNR) and multi-scale structural similarity index measure (MS-SSIM) Wang et al. (2003) measure how well the learned images learns fine details in the input. The MPE with the smallest grid cells was shown to increase the minimum eigenvalue 2 orders of magnitude over the highest frequency FFE and 8 orders of magnitude over the baseline MLP. This corresponds to an increase of 12 dB (PSNR) / 0.33 (MS-SSIM) and 15 dB (PSNR) / 0.65 (MS-SSIM), respectively.

## 2 RELATED WORK

Encodings embed a low dimensional input into a higher dimensional space before passing it to the network. Much like the kernel trick for support vector machines (Schölkopf et al., 1999), the hope is that this new space is then easier for the network to act on. A key example of this fact is found in the transformer architecture (Vaswani et al., 2017). Transformers augment their input with a positional encoding which gives the attention block information about an input's location in a larger sequence. The positional encoding updates inputs with a series of sines and cosines, providing an early example of the FFE. NeRF (Mildenhall et al., 2020) used the positional encoding as inspiration

to help a coordinate based MLP solve the inverse rendering problem, in which a scene is learned from images. Though NeRF has been iterated on, improved, and extended (Barron et al., 2021; Pumarola et al., 2020; Zhang et al., 2020), the FFE encoding has been a mainstay. The two most common FFEs are the axis-aligned logarithmic, as explored in this paper, and the Gaussian FFE (Rahimi & Recht, 2008; Tancik et al., 2020), which introduces randomness into the encoding.

Multigrid encodings, in contrast, evolved from learned grid (Fridovich-Keil et al., 2021) and voxel (Chibane et al., 2020) representations. Grid parameters, such as spherical harmonic coefficients, are learned through backpropagation and gradient descent. These representations, though intuitive, often resulted in large memory usage when capturing fine detail. So, the grid resolution was reduced, and intermediate results were passed to an MLP network to bridge the gap between the grid and finer details. This idea lead to the creation of the multigrid parametric encoding (Hadadan et al., 2021) and the hash grid encoding (Müller et al., 2022). The encodings are very similar, with the hash grid encoding being optimized for fast access on a graphics processing unit. Empirically, these encodings have outperformed FFEs in many graphics applications, such as 2D image regression and implicit surface regression. Though parametric encodings have predominantly been applied to graphics applications, the scientific machine learning community has started to take notice and the hash grid encoding has been used to solve Burgers and the Navier-Stokes equations (Huang & Alkhalifah, 2023).

The strong empirical performance of both types of encoding has sparked an interest in understanding how they work. Some authors considered empirical metrics such as gradient confusion and activation regions (Lazzari & Liu, 2023), while others have investigated the encodings through the lens of the neural tangent kernel (Tancik et al., 2020). Though gradient confusion and activation regions provide interesting insight, activation regions are restricted to ReLU activation functions, and gradient confusion is heavily dependent on the stochastic sampling in training. Therefore, we focus our analysis on the NTK. In the infinite width limit, any common machine learning architecture can be represented by a kernel regression problem using the NTK (Yang & Salman, 2019). Then, given a small enough learning rate, the training dynamics are equivalent to solving a simple ordinary differential equation (Jacot et al., 2018). This representation shows that features along the largest eigenvalues will converge faster (Basri et al., 2020; 2019; Jacot et al., 2018; Tancik et al., 2020; Wang et al., 2020). Though this kernel does not always exist in closed form, we can empirically evaluate the finite width parallel using automatic differentiation (Novak et al., 2022). Analogous to a first order Taylor expansion of the weights, the finite width kernel provides valuable NTK insights for the network sizes used in practice. Previous analysis of the NTK for coordinate based MLPs has been restricted to FFEs. We seek to include MPEs in this extensive literature, and show that the eigenvalue spectrum can be used to explain MPE's improved performance over FFEs and the baseline MLP network.

## 3 BACKGROUND AND NOTATION

In this section, we briefly explain the necessary background to understand our key results. After an overview of our notation, we describe the structure and mathematical formulation of the two most popular encodings. We then give a brief overview of the NTK and how it relates to the spectral bias problem by connecting the eigenvalues to the convergence on their corresponding eigenvector.

**Notation.** We consider a data set, $\mathbf{X} = \{\mathbf{x}_1, \mathbf{x}_2, \ldots, \mathbf{x}_N\}$ where $\mathbf{x}_i \in \mathbb{R}^d$ ; $0 < d \leq 3, i \in \{1, \ldots, N\}$. Samples are typically drawn from Cartesian space or the pixel space of an image. The corresponding target $\mathbf{Y} = \{\mathbf{y}_1, \mathbf{y}_2, \ldots, \mathbf{y}_N\}$ is similarly low dimensional. The coordinate based MLP is then given by the function $f_\theta(\mathbf{x}) : \mathbb{R}^{d_e} \to \mathbb{R}^{1,2,\text{ or }3}$ parameterized by the learnable weights $\theta$, where $d_e$ is the dimension of the embedding space. Matrices are denoted by bold capital letters, while vectors are bold lower case letters. Encodings are denoted by $\gamma(\mathbf{x}) : \mathbb{R}^d \to \mathbb{R}^{d_e}$, and the overall network is given by $f_\theta \circ \gamma$.

**Axis-Aligned Logarithmic Fourier Feature Encoding.** Axis-aligned logarithmic Fourier feature encodings are implemented by passing the original network input through a series of sines and cosines, mimicking a Fourier transform. More frequencies are added by increasing a hyperparameter $L$. Making this a learnable parameter has not been shown to work well in practice (Tancik et al., 2020). $L$ needs to be balanced with the frequency content of the function, or else the result will begin to alias and decrease in quality (Tancik et al., 2020). Written out, the encoding appears as

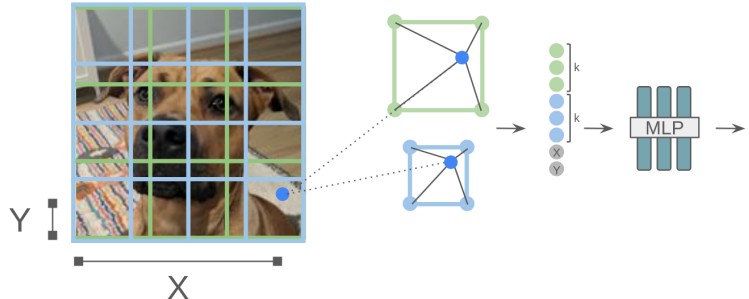

Figure 1: The above figure shows an example of the multigrid parametric encoding (MPE). The sample location (blue dot) is mapped to the surrounding grid cells (blue and green squares). The grid contains $k$ learnable scalars at each intersection point. Bilinear interpolation is performed on these learnable parameters independently. All learnable parameters are then concatenated with the origin x and y coordinate before being passed to the network.

$$\gamma_F(\mathbf{x}) = \left[ \sin(2^0 \mathbf{x}), \cos(2^0 \mathbf{x}), \sin(2^1 \mathbf{x}), \cos(2^1 \mathbf{x}), \ldots, \sin(2^{L-1} \mathbf{x}), \cos(2^{L-1} \mathbf{x}) \right]. \tag{1}$$

$\mathbf{x} \in \mathbb{R}^d$ and $\gamma_F(\mathbf{x}) \in \mathbb{R}^{d_e} = 2dL$. The logarithmic step helps the encoding shift to different frequencies due to the cyclic nature of sines and cosines. Slight variations on this encoding exist, such as phase parameters and coefficients being drawn from a standard normal distribution (Tancik et al., 2020).

**Axis-Aligned Multigrid Encoding.** Temporarily taking on the notation of Euclidean coordinates, $x$ and $y$, we consider the 2D bilinear interpolation function defined by,

$$\tilde{g}(x, y) = \frac{1}{\Delta x \Delta y} \begin{bmatrix} x_2^{(n)} - x & x - x_1^{(n)} \end{bmatrix} \begin{bmatrix} g(x_1^{(n)}, y_1^{(n)}) & g(x_1^{(n)}, y_2^{(n)}) \\ g(x_2^{(n)}, y_1^{(n)}) & g(x_2^{(n)}, y_2^{(n)}) \end{bmatrix} \begin{bmatrix} y_2^{(n)} - y \\ y - y_1^{(n)} \end{bmatrix}. \tag{2}$$

Similar functions for linear and trilinear interpolation exist for 1D and 3D applications. On an evenly spaced grid, $\Delta x$ and $\Delta y$ represent the width and height of a grid cell. The coordinates $\left[ x_1^{(n)}, y_1^{(n)} \right], \left[ x_1^{(n)}, y_2^{(n)} \right], \left[ x_2^{(n)}, y_2^{(n)} \right]$, and $\left[ x_2^{(n)}, y_1^{(n)} \right]$ represent the four corners of the $n^{th}$ cell, starting in the top left corner and moving clockwise. $g(x, y)$ is then any arbitrary function that we are interpolating.

We define a series of grids of decreasing resolution, either linearly or logarithmically, from which we interpolate the input at each coordinate. Each point in the grid contains one or more learnable scalar parameters, typically initialized by $\mathcal{N}(0, 0.01)$ (Hadadan et al., 2021). The interpolation at each resolution is then concatenated (Müller et al., 2022) along with the original input before being passed to the network. The encoding is shown graphically in Figure 1 and is given mathematically by

$$\gamma_{M,\phi}(\mathbf{x}) = \tilde{g}_\phi^{(0,0)}(\mathbf{x}) \oplus \ldots \oplus \tilde{g}_\phi^{(0,k)}(\mathbf{x}) \oplus \ldots \oplus \tilde{g}_\phi^{(L,k)}(\mathbf{x}) \oplus \mathbf{x}. \tag{3}$$

$k$ is the number of the learnable, scalar values at a given grid point, and a typical value for $k$ is in the single digits. Similarly, $L$ is the index of the grid. In practice, only a handful of layers are needed for good results. Similar encodings exist, such as the sparse multigrid encoding (Hadadan et al., 2021) and the hash grid encoding (Müller et al., 2022). Both variations work similarly and decrease the memory requirements of the encoding without changing their theoretical properties.

**Neural Tangent Kernel.** Recent work has made great strides in the explainability of neural networks using what is known as the neural tangent kernel. Jacot et al. (2018) have shown that as the width of the network tends towards infinity and the learning rate of stochastic gradient descent goes to zero, the training can be described as kernel regression with the NTK. This result can be thought

of intuitively as the Taylor expansion about the optimal weights. The NTK then represents the first order derivative in the expansion. This kernel is given by

$$\mathbf{K}_{NTK}(\mathbf{x}_i, \mathbf{x}_j) = \mathbb{E}_{\theta \sim \mathcal{N}} \left\langle \frac{\partial f_\theta(\mathbf{x}_i)}{\partial \theta}, \frac{\partial f_\theta(\mathbf{x}_j)}{\partial \theta} \right\rangle. \tag{4}$$

The kernel is formed over all pairwise comparisons within the training data set. This produces a positive semi-definite Gram matrix. With suitable initialization and infinite width, this kernel becomes deterministic and remains constant during training (Jacot et al., 2018). Assuming $K_{NTK}$ is invertible, the network's predictions on a new set of data at training step $t$ is given by (Wolberg, 2006)

$$f_\theta(\mathbf{X}_{test}, t) \approx \mathbf{K}_{test} \mathbf{K}_{NTK}^{-1} (\mathbf{I} - e^{-\mathbf{K}_{NTK}t}) \mathbf{Y}. \tag{5}$$

$\mathbf{K}_{test}$ is given by the neural tangent kernel of the pairwise inner products of the training and test data set, and $\mathbf{Y}$ is the concatenated ground truth training data set as described in the notation section. Though originally formulated in the infinite width setting, the finite width evaluation of the NTK has been a reliable performance predictor of small and medium sized neural networks (Tancik et al., 2020).

**Spectral Bias.** The NTK allows a more classical understanding of the network. $\mathbf{K}_{NTK}$ is positive semi-definite. Therefore, it can be decomposed into an orthogonal matrix and a diagonal matrix containing the eigenvalues $\lambda$ such that $\mathbf{K}_{NTK} = \mathbf{Q}^T \mathbf{\Lambda} \mathbf{Q}$. Positive semi-definiteness means that all $\lambda$'s are greater than or equal to 0. We compare the predictions of the training data by evaluating $f_\theta(\mathbf{X}, t) - \mathbf{Y}$. Multiplying the NTK and its inverse produces the identity matrix, giving

$$f_\theta(\mathbf{X}, t) - \mathbf{Y} = (\mathbf{I} - e^{-\mathbf{K}_{NTK}t}) \mathbf{Y} - \mathbf{Y}. \tag{6}$$

Using the fact that $e^{-\mathbf{K}_{NTK}t} = \mathbf{Q}^T e^{-\mathbf{\Lambda}t} \mathbf{Q}$, our training loss simplifies to

$$|\mathbf{Q}(f_\theta(\mathbf{X}, t) - \mathbf{Y})| = |-e^{-\mathbf{\Lambda}t} \mathbf{Q} \mathbf{Y}|. \tag{7}$$

From Equation 7 we see that the loss has a dependence on the eigenvalues and eigenvectors of the NTK and that larger eigenvalues will decrease the training error faster along that dimension. Basri et al. (2019) found that eigenvalues corresponding to higher frequency relationships in the dataset are lower for coordinate-based MLPs. As a result, networks will fit lower frequency, smooth, relationships much faster than they will the higher frequencies. By comparing the eigenvalue spectrum produced by the composition of encodings and the NTK, we can evaluate an encodings' bias towards higher frequency features, which corresponds to an increased number of large eigenvalues.

## 4 NEURAL TANGENT KERNELS FOR MULTIGRID PARAMETRIC ENCODINGS

We explore the structure of the neural tangent kernel for different encodings and prove that the MPE raises the eigenvalue spectrum of the kernel over the baseline network. In special cases, the NTK can be written out in closed form for its infinite width limit. We, however, focus on the finite width kernel to evaluate training. By extending previous work on the NTK of MLP networks (Jacot et al., 2018; Yang & Salman, 2019), we derive the kernel for MPEs and FFEs. We start by analyzing a single layer MLP network, extend it to include encodings, and discuss how it is easily extended for MLPs with multiple layers.

Consider a single layer network without encoding and width $n$. Taking on the parameterization scheme in Jacot et al. (2018), we scale the activation by the dimension $n$ and introduce a scalar, $\beta$, to scale the bias. $\beta$ is typically set to 0.1. Given weights, $\mathbf{W}^{(1)} \in \mathbb{R}^{n \times d}$ and $\mathbf{W}^{(2)} \in \mathbb{R}^{1 \times n}$, and biases, $\mathbf{b} \in \mathbb{R}^{n \times 1}$, we consider the function

$$f_\theta(\mathbf{x}) = \mathbf{W}^{(2)} \frac{1}{\sqrt{n}} \phi(\mathbf{W}^{(1)} \mathbf{x} + \beta \mathbf{b}). \tag{8}$$

Viewing $\theta$ as the concatenation of all learnable parameters, such that $\theta = \begin{bmatrix} \mathbf{W}^{(2)} & \mathbf{W}^{(1)} & \mathbf{b} \end{bmatrix}$, we take the derivative with respect to each parameter matrix. The $i, j$ element of $K_{NTK}$ is then given by the inner product between $\mathbf{x}_i$ and $\mathbf{x}_j$ in the data set. Therefore,

$$
\begin{aligned}
\mathbf{K}_{NTK}^{(i,j)} = \mathbb{E}_{\theta \sim \mathcal{N}} \langle \frac{\partial f_\theta(\mathbf{x}_i)}{\partial \theta}, \frac{\partial f_\theta(\mathbf{x}_j)}{\partial \theta} \rangle = \mathbb{E}_{\theta \sim \mathcal{N}} \big[ \frac{1}{n} \langle \phi(\mathbf{W}^{(1)}\mathbf{x}_i + \beta\mathbf{b}), \phi(\mathbf{W}^{(1)}\mathbf{x}_j + \beta\mathbf{b}) \rangle \\
+ \frac{1}{n} \langle \mathbf{W}^{(2)}\mathbf{x}_i \frac{\partial \phi}{\partial \theta}(\mathbf{W}^{(1)}\mathbf{x}_i + \beta\mathbf{b}), \mathbf{W}^{(2)}\mathbf{x}_j \frac{\partial \phi}{\partial \theta}(\mathbf{W}^{(1)}\mathbf{x}_j + \beta\mathbf{b}) \rangle \\
+ \frac{\beta^2}{n} \langle \mathbf{W}^{(2)} \frac{\partial \phi}{\partial \theta}(\mathbf{W}^{(1)}\mathbf{x}_i + \beta\mathbf{b}), \mathbf{W}^{(2)} \frac{\partial \phi}{\partial \theta}(\mathbf{W}^{(1)}\mathbf{x}_j + \beta\mathbf{b}) \rangle \big].
\end{aligned}
\quad (9)
$$

We now include the multigrid parametric encoding. As gradients are computed element wise across the training dataset, we only need to consider a single grid cell, layer, and learnable scalar at a time. Using the notation from section 3.3, define $\Delta x_2 = x_2^{(n)} - x, \Delta x_1 = x - x_1^{(n)}, \Delta y_2 = y_2^{(n)} - y$, and $\Delta y_1 = y - y_1^{(n)}$. We drop the $n$ superscript to consider a single cell, with learnable weights $w_{11}, w_{12}, w_{21}$, and $w_{22}$. Writing out the matrices and multiplying through, we get

$$
\begin{bmatrix} \Delta x_2 & \Delta x_1 \end{bmatrix} \begin{bmatrix} w_{11} & w_{12} \\ w_{21} & w_{22} \end{bmatrix} \begin{bmatrix} \Delta y_2 \\ \Delta y_1 \end{bmatrix} = w_{11}\Delta y_2 \Delta x_2 + w_{12}\Delta y_1 \Delta x_2 + w_{21}\Delta y_2 \Delta x_1 + w_{22}\Delta y_1 \Delta x_1.
$$
$$(10)$$

It should be apparent that the gradient with respect to the weights in the grid cell is

$$
\nabla_{\mathbf{W}} \tilde{g}(x, y) = \tilde{g}'(x, y) = \frac{1}{\Delta x \Delta y} \begin{bmatrix} \Delta x_2 \\ \Delta x_1 \end{bmatrix} \begin{bmatrix} \Delta y_2 & \Delta y_1 \end{bmatrix}.
\quad (11)
$$

The new function is the composition of $f_\theta$ with the encoding, $\gamma_\theta$. As before, we concatenate all learnable parameters and compute the $i^{th}, j^{th}$ index of $K_{NTK}$. The gradients of the grid parameters are independent due to the concatenation, so we first compute the contribution of a single grid parameter. The full kernel is then the sum of each grid cell contribution. Let $\tilde{g}(\mathbf{x}_i)$ be the output of the bilinear interpolation on a single grid cell, then the kernel element is

$$
\begin{aligned}
\mathbf{K}_{NTK_{MPE}}^{(i,j)} = \mathbb{E}_{\theta \sim \mathcal{N}} \langle \frac{\partial f_\theta(\gamma_\theta(\mathbf{x}_i))}{\partial \theta}, \frac{\partial f_\theta(\gamma_\theta(\mathbf{x}_j))}{\partial \theta} \rangle = \\
\mathbb{E}_{\theta \sim \mathcal{N}} \big[ \frac{1}{n} \langle \phi(\mathbf{W}^{(1)}\tilde{g}(\mathbf{x}_i) + \beta\mathbf{b}), \phi(\mathbf{W}^{(1)}\tilde{g}(\mathbf{x}_j) + \beta\mathbf{b}) \rangle \\
+ \frac{1}{n} \langle \mathbf{W}^{(2)}\tilde{g}(\mathbf{x}_i) \frac{\partial \phi}{\partial \theta}(\mathbf{W}^{(1)}\tilde{g}(\mathbf{x}_i) + \beta\mathbf{b}), \mathbf{W}^{(2)}\tilde{g}(\mathbf{x}_j) \frac{\partial \phi}{\partial \theta}(\mathbf{W}^{(1)}\tilde{g}(\mathbf{x}_j) + \beta\mathbf{b}) \rangle \\
+ \frac{\beta^2}{n} \langle \mathbf{W}^{(2)} \frac{\partial \phi}{\partial \theta}(\mathbf{W}^{(1)}\tilde{g}(\mathbf{x}_i) + \beta\mathbf{b}), \mathbf{W}^{(2)} \frac{\partial \phi}{\partial \theta}(\mathbf{W}^{(1)}\tilde{g}(\mathbf{x}_j) + \beta\mathbf{b}) \rangle \\
+ \frac{1}{n} \langle \mathbf{W}^{(2)}\mathbf{W}^{(1)}\tilde{g}'(\mathbf{x}_i) \frac{\partial \phi}{\partial \theta}(\mathbf{W}^{(1)}\tilde{g}(\mathbf{x}_i) + \beta\mathbf{b}), \mathbf{W}^{(2)}\mathbf{W}^{(1)}\tilde{g}'(\mathbf{x}_j) \frac{\partial \phi}{\partial \theta}(\mathbf{W}^{(1)}\tilde{g}(\mathbf{x}_j) + \beta\mathbf{b}) \rangle \big].
\end{aligned}
\quad (12)
$$

The first three terms correspond to the original MLP with the input in the embedding space of the MPE instead of the original coordinate space. Let's denote this with $\mathbf{K}_{MLP}^{i,j}$. The last term is a new term induced by the parameters in the grid, which we'll label $\mathbf{K}_{MPE}^{i,j}$. As layers along the $L$ dimension and trainable parameters along the $k$ dimension are independent, their kernel contribution is simply Equation 12 repeated for each layer and parameter plus the kernel evaluated on the original coordinates. Expanding out across all elements $\mathbf{x}_i \in \mathbf{X}$, we get the following kernel for the MPE:

$$
\mathbf{K}_{NTK_{MPE}} = \mathbf{K}_{NTK} + \sum_{l=1}^{L} \sum_{a=1}^{k} \mathbf{K}_{MLP}^{l,a} + \sum_{l=1}^{L} \sum_{a=1}^{k} \mathbf{K}_{MPE}^{l,a}.
\quad (13)
$$

The NTK induced by the composition of the MPE and a MLP is the sum of the original NTK, Equation 9, without any encoding and the NTK produced by each layer and learnable parameter, Equation 12. Though we only discuss a single layer MLP to illustrate the structure of the NTK, the extension to multiple layers is well established and can be built recursively from the above definitions (Jacot et al., 2018; Yang & Salman, 2019). Adding layers to the MLP will simply add terms to Equation 9 and Equation 12. Equation 13, however, still holds for deeper MLPs. We use this fact to prove that the eigenvalue spectrum of the composed kernel is greater than that of the base MLP.

**Theorem 1.** *Given a dataset* **X***, with $n$ samples, the corresponding neural tangent kernel for a MLP network, and the neural tangent kernel for the same dataset and MLP composed with a MPE. The $i^{th}$ eigenvalue, sorted in descending order, of each kernel follows $\lambda_i^{MLP} \leq \lambda_i^{MLP} + \lambda_n^{MPE} \leq \lambda_i^{MLP+MPE}; \forall i \in \{1, \ldots, n\}$. $\lambda_i^{MLP}$ are the eigenvalues of the NTK for the network with no encoding. $\lambda_n^{MPE}$ is the smallest eigenvalue of the matrix produced by the encoding layers shown in Equation 13. $\lambda_i^{MLP+MPE}$ are then the eigenvalues of the NTK for the encoded network.*

*Proof.* Let $\mathbf{K}_{MLP}$ by the neural tangent kernel for a MLP network evaluated on a training dataset $\mathbf{X}^n$. Let $\mathbf{K}_{MPE}$ be the neural tangent kernel for the composed MLP and MPE evaluated on the same dataset. From Equation 13 we see that $\mathbf{K}_{MPE} = \mathbf{K}_{MLP} + \mathbf{K}^+$, where $\mathbf{K}^+$ is the kernel produced by the sum over all learnable parameters in the grid. By construction, each $\mathbf{K}$ are square, symmetric Gram matrices as they are constructed by the inner product. Each $\mathbf{K}$ is, therefore, positive semidefinite, making $\mathbf{K}_{MPE}$ the sum of positive semidefinite matrices. Let $\lambda_i(\mathbf{K})$ denote the $i^{th}$ eigenvalue of the matrix $\mathbf{K}$. It follows from Weyl's inequality (Weyl, 1912) that $\forall i \in \{1, \ldots, n\}, \lambda_i(\mathbf{K}_{MLP}) \leq \lambda_i(\mathbf{K}_{MLP}) + \lambda_n(\mathbf{K}^+) \leq \lambda_i(\mathbf{K}_{MPE})$. □

We see that the MPE changes the kernel both through its embedding space and the learnable parameters contained within the grid. By adding more layers in the grid or more learnable parameters at the grid nodes, we can increase the spectrum of the kernel at the cost of additional computation and memory usage. It is possible that the minimum eigenvalue of the additional matrices is zero; however, as will be demonstrated in the next section, this is far from true in practice.

For comparison, the NTK for the FFE can be computed as

$$
\mathbf{K}_{NTK_{FFE}}^{(i,j)} = \mathbb{E}_{\theta \sim \mathcal{N}} \langle \frac{\partial f_\theta(\gamma_F(\mathbf{x}_i))}{\partial \theta}, \frac{\partial f_\theta(\gamma_F(\mathbf{x}_j))}{\partial \theta} \rangle =
$$

$$
\mathbb{E}_{\theta \sim \mathcal{N}} \big[ \frac{1}{n} \langle \phi(\mathbf{W}^{(1)}\gamma_F(\mathbf{x}_i) + \beta\mathbf{b}), \phi(\mathbf{W}^{(1)}\gamma_F(\mathbf{x}_j) + \beta\mathbf{b}) \rangle
$$

$$
+ \frac{1}{n} \langle \mathbf{W}^{(2)}\gamma_F(\mathbf{x}_i) \frac{\partial\phi}{\partial\theta}(\mathbf{W}^{(1)}\gamma_F(\mathbf{x}_i) + \beta\mathbf{b}), \mathbf{W}^{(2)}\gamma_F(\mathbf{x}_j) \frac{\partial\phi}{\partial\theta}(\mathbf{W}^{(1)}\gamma_F(\mathbf{x}_j) + \beta\mathbf{b}) \rangle
$$

$$
+ \frac{\beta^2}{n} \langle \mathbf{W}^{(2)} \frac{\partial\phi}{\partial\theta}(\mathbf{W}^{(1)}\gamma_F(\mathbf{x}_i) + \beta\mathbf{b}), \mathbf{W}^{(2)} \frac{\partial\phi}{\partial\theta}(\mathbf{W}^{(1)}\gamma_F(\mathbf{x}_j) + \beta\mathbf{b}) \rangle \big]. \quad (14)
$$

Equation 14 shows that the FFE improves the NTK solely through its embedding space, and Theorem 1 does not apply. The MPE, in contrast, has two mechanism to influence the kernel, but which is dominant? To isolate the improvements in the MPE, we compute the spectrum both with and without the $\mathbf{K}_{MPE}$ term (Figure 2). Without the contributions of the learnable grid, the MPE has little to no effect on the eigenvalues as compared to baseline. We conclude that the MPE derives its performance from the learnable parameters and not the higher dimensional embedding space, while the FFE's performance rests solely on the embedding.

## 5 EXPERIMENTS AND RESULTS

We present an empirical analysis of the neural tangent kernel for multiple configuration of the FFE and MPE in the context of a 2D image regression and a 3D implicit surface regression problem. Additional results can be found in Appendix A. The image regression problem provides a controlled setting that is easily interpretable, allowing us to understand how the theoretical properties of the kernel translate to empirical results, considering convergence rates, prediction accuracy, and sensitivity to hyperparameters. 3D implicit surface regression demonstrates that the theory holds in

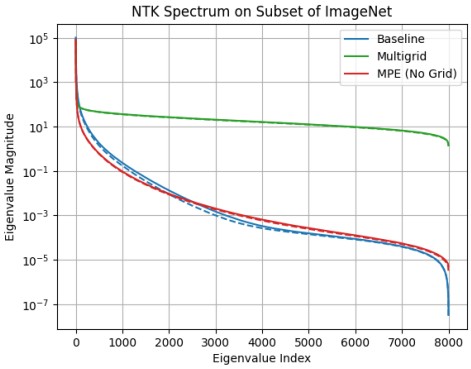

Figure 2: This plot isolates the improvements of the MPE to the learnable grid. The NTK is computed for a 2D image regression on random images from 100 synonym sets in ImageNet (See the *Experiments and Results* section for more details). The solid line and dashed line are the spectrum at the end and the middle of training, respectively. MPE (No Grid) is the NTK of the MPE without the contributions of $K_{MPE}$ and is purely for theoretical analysis. Without the grid, the spectrum is barely above the baseline. With the grid, the spectrum is 8 magnitudes higher.

higher dimensions as well, and provides a theoretical basis for previous results by Müller et al. (2022).

**Experimental Setup.** We evaluate the NTK in three settings. The first setting explores the relationship between the eigenvalue spectrum and the encoding hyperparameters on a single image. The second setting extends our results to a larger class of images by evaluating the NTK on randomly sampled images from 100 synonym sets from ImageNet (Deng et al., 2009). The final setting demonstrates that the theory holds when learning a 3D implicit surface from a mesh. The network size was held constant in each domain. For image regression, 2 hidden layers with 512 neurons each were used, while the 3D surface regression used 8 hidden layers with 256 neurons each. The ReLU activation function was use in intermediate layers. For more details on 2D regression, please see Appendix A and see Appendix E for 3D implicit surface information. The baseline corresponds to the network with no encoding, while the FFE and MPE hyperparameters were set as follows. The scaling experiment hyperparameters (Table 3) were hand selected, while the ImageNet hyperparameters (Table 2) and 3D implicit surface hyperparameters (Table 4) were selected using Optuna (Akiba et al., 2019).

**Evaluating the NTK and Regression.** We used automatic differentiation and the fast finite width NTK calculation (Novak et al., 2022) to evaluate the kernel during training. The eigenvalues of the kernel were computed at the beginning, middle, and end of training. The eigenvalues are then sorted from high to low and plotted on a logarithmic scale. To evaluate the regressed image, we report both peak signal-to-noise ratio (PSNR) in dB and multi-scale structural similarity index measure (MS-SSIM) Wang et al. (2003). PSNR reports the max signal power as compared to the noise (average error); however, as PSNR is an MSE based metric, it may not capture fine detail, so we also include MS-SSIM which is more sensitive to the fine feature of the image. Higher scores in both metrics corresponds to higher quality outputs from the network as compared to ground truth.

**Scaling Analysis.** Figure 3 shows the regressed image at the end of training. Qualitatively, we can see the spectral bias and how the encodings mitigate it. The baseline is only able to learn a blurred image. The low frequency FFE greatly improves the results but still lacks sharpness. This fine detail is learned more easily by the higher frequency FFEs. Both MPEs easily learn high frequency details without greatly increasing the network's input size. This qualitative analysis is corroborated by the PSNR and MS-SSIM values in Table 3.

Figure 4 plots the corresponding eigenvalue spectrum for each of these regression problems shown in Figure 3. The eigenvalue spectrum at the middle and end of training are plotted as the dashed and solid lines, respectively. The baseline encoding has the worst performance, with the eigenvalues quickly dipping below $1e - 6$. The left most plot shows the comparison against all three FFEs. As

| Metric | ImageNet | | | Scaling | | | | | |
|---|---|---|---|---|---|---|---|---|---|
| | Multigrid | Fourier | Baseline | Low | Mid | High | Coarse | Fine | Baseline |
| PSNR ↑ | 45.28 | 33.47 | 29.94 | 25 | 35 | 36.5 | 34.5 | 41.5 | 20 |
| m-SSIM ↑ | 0.79 | 0.39 | 0.32 | 0.20 | 0.23 | 0.40 | 0.37 | 0.73 | 0.08 |

Table 1: This table presents image quality metrics between the regressed image and ground truth across experiments. PSNR and MS-SSIM evaluate this similarity, with the latter being more sensitive to fine detail. As expected, we see that the MPE has the highest scores, while the FFE and MPE perform well over the baseline. See Figure 3 for a visual representation of these trends.

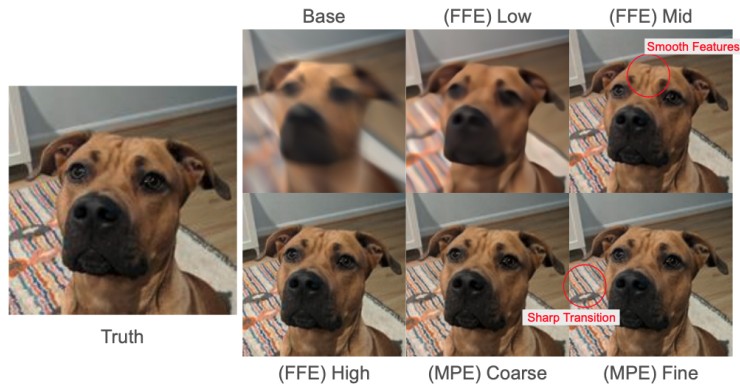

Figure 3: We compare performance of different encodings on image regression. We show the ground truth image (leftmost) along with a network with no encoding (top left), 3 configurations of the Fourier feature encoding (FFE), and two configurations of the multigrid parametric encoding (MPE) (see Table 3). No encoding produces a blurred image, but as we add encodings, the finer details start to be resolved. An increase in detail is seen in the FFEs, while the MPEs perform well with both the coarse and the fine grid.

expected, increasing frequency corresponds to an increase in the eigenvalues; however, we see that the FFE saturates, and there is little improvement between the mid and high frequencies. The middle figure plots the same comparison for the MPEs. Again, as theory predicts, the higher performing fine MPE has a higher eigenvalue spectrum than the coarse grid. The right figure then compares the two MPEs to the high frequency FFE. The fine grid has the highest spectrum overall, while the FFE and coarse MPE flip around the middle of the plot. This could explain why the PSNR and MS-SSIM values for the two encodings are so similar. The coarse MPE will quickly learn finer detail than the baseline, but the FFE will be able to learn the very fine details at the smallest eigenvalues. Appendix A shows the same results across encodings on additional images.

**ImageNet Analysis.** Plots of the average eigenvalue spectrum for the tuned encodings on ImageNet are found in Figure 5a. Again, solid lines and dashed lines are the spectra at the end and middle of training, respectively. As images could have different numbers of pixels, the spectra were scaled to 8000 values. The mean was computed for each category. Tuning reflects the use of encodings in practice, and we find that the MPE shows clear benefits in the spectrum on a wide variety of images. The clear delineations of the spectra also demonstrates the robustness of this evaluation on a wide class of images and strong alignment between the performance of the network and the spectrum of the NTK.

**OccupancyNet Analysis.** Lastly, we plot the average eigenvalue spectrum for evaluations across three 3D meshes from the Stanford graphics dataset Turk & Levoy (1994) in Figure 5b. Again, we see a clear benefit of the MPE, with the spectra raising above the other encodings. The minimum eigenvalues of the FFE and MPE are similar, but the overall spectrum for the MPE is higher. This corresponds to better training across all eigenvectors. Both encodings show a clear improvement over the baseline, as expected. Visualization of the 3D surface are in Appendix E. These results show that the theory easily extends from 2D to 3D, and that the NTK spectrum is a powerful tool in understanding the performance of encodings across domains.

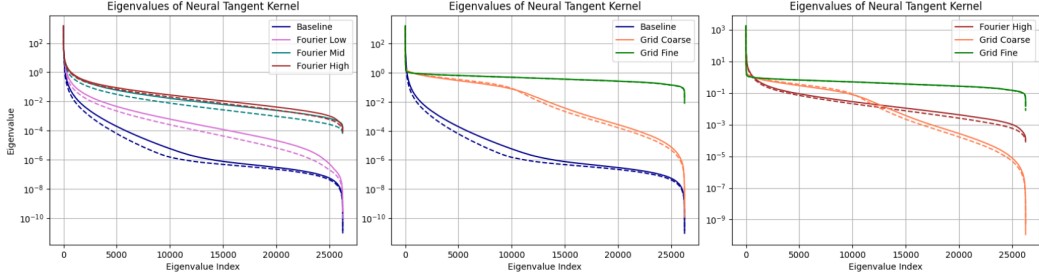

Figure 4: The NTK eigenvalue spectrum is compared for the cases found in Figure 3 and Table 3. All encodings perform better than the baseline. The left plot shows the comparison of the baseline and the FFEs. The middle plot shows the same for the baseline and the MPEs. The right blot then compares the high frequency FFE to both MPE. The FFEs seem to saturate, while the fine grid MPE gives the best performance. These trends are backed by the PSNR values reported in the table. Interestingly, the coarse MPE crosses over the FFE, giving it strong performance early in training but allowing for higher PSNR values in the FFEs at the end of training.

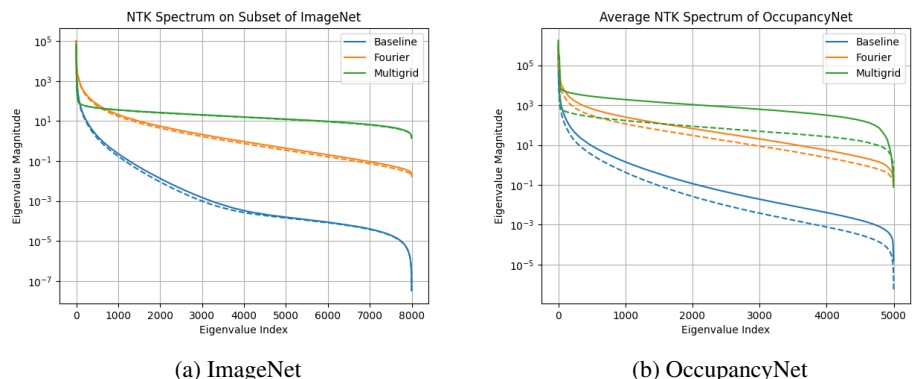

| (a) ImageNet | (b) OccupancyNet |

Figure 5: This figure compares the mean of the eigenvalue spectra of different encodings across randomly sampled images from 100 synonym sets in ImageNet and three 3D meshes on OccupancyNet. The dashed line shows the mean spectra at the midpoint of training. We find that tuned encodings have fairly regular performance: the MPE outperforms the FFE, which outperforms the baseline. This result shows that the spectrum is stable across different images and domains.

## 6 CONCLUSION, LIMITATIONS, AND FUTURE WORK

We present an analysis of the multigrid parametric encoding compared to the Fourier feature encoding and baseline coordinate-based MLP to understand the MPE's strong empirical performance in many graphics applications. We follow previous analyses on coordinate based MLPs for graphics and scientific machine learning and leverage neural tangent kernel theory to connect the eigenvalue spectrum with the learning of higher frequency information. We prove a lower bound on the eigenvalues of the MPE and give strong empirical evidence for this improvement being isolated to the learnable grid as opposed to the embedding space. The MPE, therefore, mitigates spectral bias in a fundamentally different way than the FFE. Our analysis, however, does not account for the influence of different activation functions, and the space of possible problems is too large to compute the NTK for each application. Evaluations on a 2D image regression and 3D implicit surface problem demonstrated that the theory holds in multiple domains, while providing easily interpreted visual results. Through metrics such as PSNR and MS-SSIM, we showed strong alignment between the theory and empirical results. Our work greatly improves the community's understanding of encodings and could be leveraged in future works to better optimize them to specific applications by adjusting the interpolation kernel or tuning other hyperparameters.

ACKNOWLEDGMENTS

This project was supported in part by a grant from an NSF CAREER AWARD 1942230, ONR YIP award N00014-22-1-2271, ARO's Early Career Program Award 310902-00001, Army Grant No. W911NF2120076, the NSF award CCF2212458, NSF Award No. 2229885 (NSF Institute for Trustworthy AI in Law and Society, TRAILS), a MURI grant 14262683, an award from meta 314593-00001, and an award from Capital One.

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

## A  TRAINING DETAILS AND FURTHER EXPERIMENTS

### A.1  TRAINING DETAILS

The PyTorch (Paszke et al., 2019) library was used to implement all models and analyses. Runs were completed on internal grid infrastructure using a single NVIDIA RTX A5000 graphics card, 8 CPU cores, and 56 GB of system memory. The GPU handles training and computation of the neural tangent kernel, while the CPU handles the eigenvalue decomposition, as it was found to be faster. Each analysis takes approximately 30 minutes to complete.

2D image regression trains the network on a single image. The pixel coordinates and red, green, and blue color values were normalized to between 0 and 1. There was no split between training and test data because, even with the complete dataset, the coordinate based MLP is not able to learn the high frequency information in the image. Training was done using stochastic gradient descent Kiefer & Wolfowitz (1952) with a mean squared error loss between the linear activation of the final layer and the ground truth pixel value.

To evaluate the NTK during training, slight variations are made to the network architecture. NTK initialization (Jacot et al., 2018) of the linear layers was used in which the weights and biases are sampled from $\mathcal{N}(0, \mathbf{I})$ and a layer, $l$, is given by $\sigma(\frac{1}{\sqrt{n^{(l)}}}\mathbf{W}^{(l)}\mathbf{x}^{(l)} + \beta\mathbf{b}^{(l)})$, where $n^{(l)}$ is the size of the input vector, $x^{(l)}$, to that layer. Though this parameterization has the same approximation power as the standard MLP, the additional factor greatly reduces the gradients during backpropagation, requiring us to set the learning rate to 100 on some experiments.

### A.2  DOG IMAGE

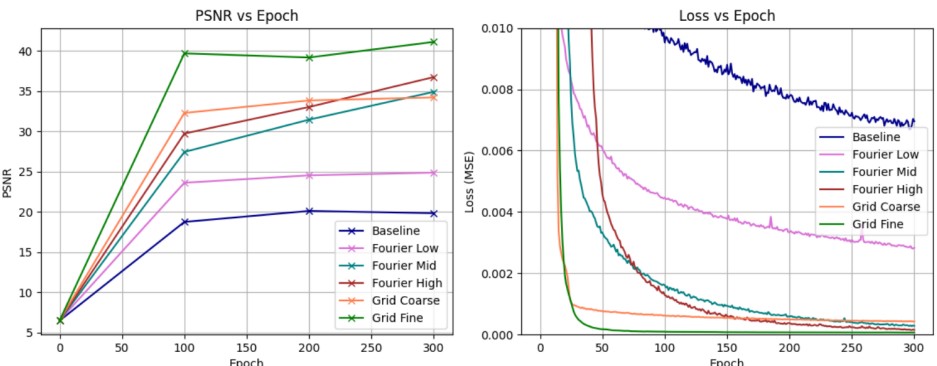

Figure 6: We report the peak signal-to-noise ratio (PSNR) and mean squared error (MSE) for the image regression problem shown in Figure 3. We compare a baseline network with no encoding, 3 Fourier feature encoding (FFE), and 2 multigrid parametric encoding (MPE). These results parallel what was seen in the regressed images. The MSE is highest and PSNR is lowest for the baseline encoding corresponding to the lowest quality image. As frequencies increase in the FFE, we see the MSE lower and the PSNR increase. The fine MPE outperforms even the highest frequency FFE by 5 dB PSNR, while the coarse MPE is on par with the higher frequency FFEs. We also note that the MPE lowers the error at a faster rate than the FFE, with loss dropping below 1e-3 125 epochs sooner.

Figure 6 compares the MSE loss and PSNR across encodings. As expected, the baseline network has the highest training error and the lowest PSNR. We find that these plots follow the qualitative results from Figure 3. The low frequency encoding halves MSE and increases the PSNR by 5 dB. The mid and high frequency encodings show similar results in both the regressed image and MSE and PSNR. The higher frequency encodings halve the loss again and increase the PSNR by 10 dB as compared to the low frequency encoding. The coarse multigrid encoding results in similar MSE and PSNR compared to the mid and high frequency encodings; however, it trains much faster with the loss dropping below 1e-3 125 epochs sooner than the mid and high frequency encodings. The best performing encoding is then the fine multigrid encoding with a PSNR another 5 dB higher than the

coarse MPE, mid PPE, and high PPE. Loss is also the lowest for the fine MPE. From these results, NTK theory would tell us that the kernel's eigenvalues would follow the same trend.

## A.3 WINDMILL AND LAKE IMAGES

We include results on two other images for the 2D image regression problem across encodings. We show that the same results from the main paper hold across multiple data sets. We include both a natural lake scene and a combination man made and natural scene featuring a windmill and flowerbed. Both a qualitative look at the regressed images and MSE and PSNR show the same trends. The base network is unable to learn fine details in the image. The three FFEs show improving results with increasing frequency. The fine grid encoding then has the best performance with the lowest training error, the largest PSNR, and the highest eigenvalue spectrum.

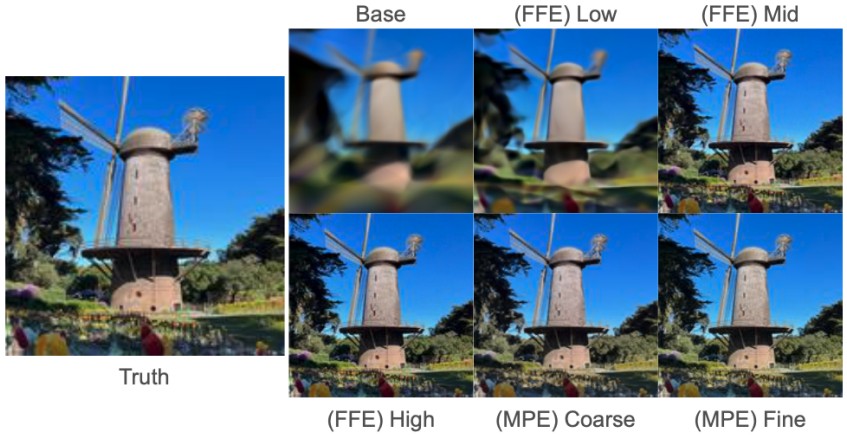

Figure 7: Results for the 2D image problem on an image of a windmill in a natural setting. See Section 5 for details on training and the parameters of each encoding. With no encoding (top left), the regressed image is blurry with no fine, high frequency details. The low frequency (top middle) encoding learns slightly more detail, but still results in a blocky image. The mid frequency encoding (top right), begins to show strong agreement with the ground truth image. The high frequency (bottom left), coarse grid (bottom middle), and fine grid (bottom right) encodings show even stronger agreement with the ground truth image, to the point where it is difficult to tell the difference with the human eye.

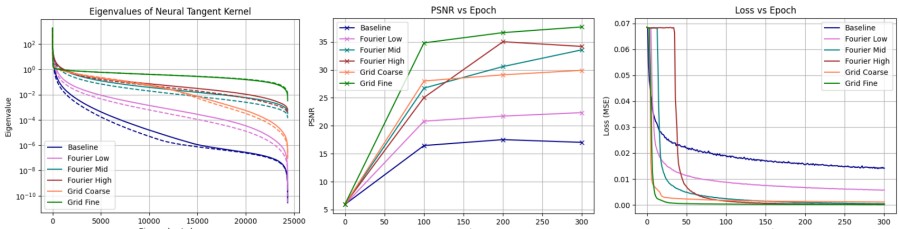

Figure 8: We plot the eigenvalue spectrum (left) at epoch 150 (dashed line) and epoch 300 (solid) line, the PSNR throughout training (middle), and the loss curves (right) to compare the performance of the encodings quantitatively on the windmill image. Similarly to the main paper, we find that the strongest performing encodings, highest PSNR and lowest training loss, has the highest magnitude eigenvalues spectrum. Lower eigenvalue spectra then correspond to lower PSNR and higher loss throughout training.

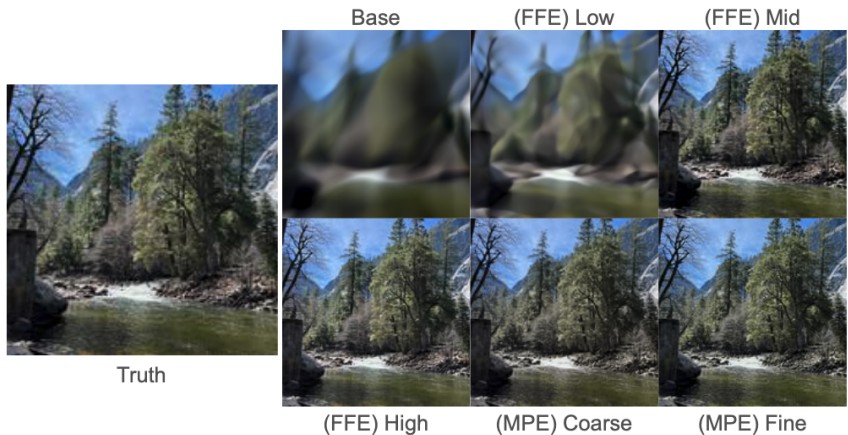

Figure 9: Results for the 2D image problem on an image of a lake in a natural setting. See Section 5 for details on training and the parameters of each encoding. With no encoding (top left), the regressed image is blurry with no fine, high frequency details. The low frequency (top middle) encoding learns slightly more detail, but still results in a blocky image. The mid-frequency encoding (top right), begins to show strong agreement with the ground truth image. The high frequency (bottom left), coarse grid (bottom middle), and fine grid (bottom right) encodings show even stronger agreement with the ground truth image, to the point where it is difficult to tell the difference with the human eye.

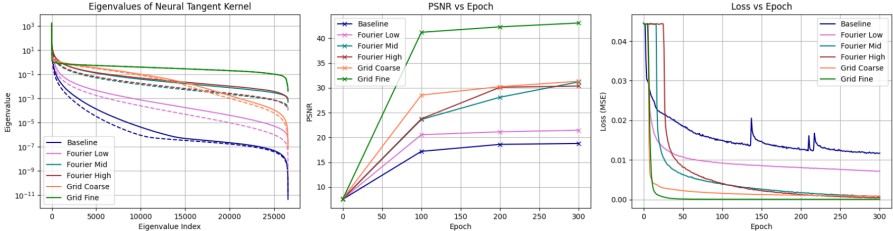

Figure 10: We plot the eigenvalue spectrum (left) at epoch 150 (dashed line) and epoch 300 (solid) line, the PSNR throughout training (middle), and the loss curves (right) to compare the performance of the encodings quantitatively on the lake image. Similarly to the main paper, we find that the strongest performing encodings, highest PSNR and lowest training loss, has the highest magnitude eigenvalues spectrum. Lower eigenvalue spectra then correspond to lower PSNR and higher loss throughout training.

## B    CHANGE OF NTK DURING TRAINING

When plotting the eigenvalue spectrum, we only show results at epoch 150 and epoch 300. In Figure 11 we show the results for epoch 0 and epoch 300. As shown, the spectrum at the start of training is very different from the spectrum at the end of training. We also note that eigenvalues across encodings all seem to have similar values. NTK theory says that the kernel is stable in the infinite width limit (Jacot et al., 2018), and a width of 512 should be large enough to see this effect. So why does that not hold in this case? Recent works by Wang et al. (2021b; 2020) have shown that the eigenvalue spectrum does, in fact, change during training in real world scenarios. The reason given is that the initial initialization is too far from the optimal parameter values, so large changes in weights are still necessary for learning. This is most likely exacerbated by the fact our inputs are scaled to the unit interval and not a unit normal distribution. This breaks the lazy training assumption for the infinite width limit of the kernel; however, we are focused on the finite width kernel and the training dynamics given different encodings. Furthermore, the spectral bias analysis using the eigenvalues of the NTK still holds. This can be seen by considering the Taylor expansion about the optimal weights. Given slight perturbations, we expect the kernel to be elatively stable in this region, and its

eigenvalues allow us to conclude with eigenvectors will be fit faster during training. This is reflected in the fact that the kernel spectrum is stable after the start of training, as shown in the main Figures.

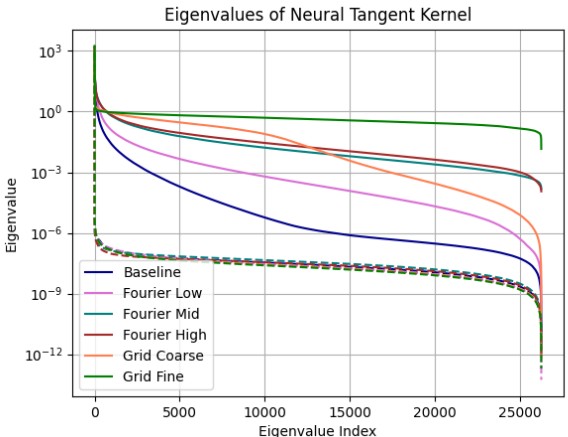

Figure 11: We plot the eigenvalues at the start of training (dotted line) and at epoch 300 (solid line). We see that the spectrum is very low and close between encodings at the start of training, but raises and separates by the end of training. Though this appears to break the lazy training observation of the infinite width kernel (Jacot et al., 2018), we are concerned with the finite width kernel and the spectral bias analysis still holds as long as we consider small regions about the weights at a time. As show in Figure 4, the kernel is relatively constant for the second half of training.

## C  VISUALIZING MULTIGRID PARAMETERS

Though we have shown that the multigrid encoding is able to raise the eigenvalue spectrum and expressiveness of a coordinate based MLP, it is also interesting to look at what the parameters in the grid are learning. In Figure 12 we plot the parameters as a gray scale image for both the coarse and fine encodings. Each encoding contains two learnable scalars at each grid point. The $0^{th}$ parameter is shown on the left while the $1^{st}$ is plotted on the right. We see that the grid, in fact, begins to learn the image, while the backing MLP adjusts the color and helps interpolate. We also see small gaps in the fine encoding, most likely due to grid cells that were unused during training. This shows that it's easy to waste space, especially with uniform sampling. In these cases, the sparse and hash grid encodings could help reduce the memory footprint without changing the analyses found in the main paper.

If the multigrid encoding is learning the image, then does the MLP need to do less work? Put another way, can we reduce the size of the MLP to save memory and computation. To investigate this question, we plot the activation regions (Hanin & Rolnick, 2019) in Figure 13. Activation regions encode which neurons in the MLP cause the ReLU activation function to take on a value of 1. This produces a binary encoding that can be used to define an activation pattern, $\mathcal{A}$, which is a vector in $\{0, 1\}^{\#neurons}$. Each unique activation pattern then defines an activation region by

$$\mathcal{R}(\mathcal{A}, \theta) = \{x \in \mathbb{R}^{\dim(x)} \mid \chi_+(\text{ReLU}(f^l(x_l))) = a_l, \; \forall l, \ldots, k, \; \forall a_l \in \mathcal{A}\}. \tag{15}$$

Plotting the activation region for each pixel then shows the different sets of activations present in the network. The closer the image looks to noise, the more activation patterns are present. In Figure 13 we show both the activation regions across the unit square and the total number of activation regions present across the different encodings. We find that the network has similar activation for both the MPEs and higher frequency FFEs. As such, we believe that the network is still playing a large role in the architecture, and we should be cautious about reducing its capacity.

Fine    Coarse

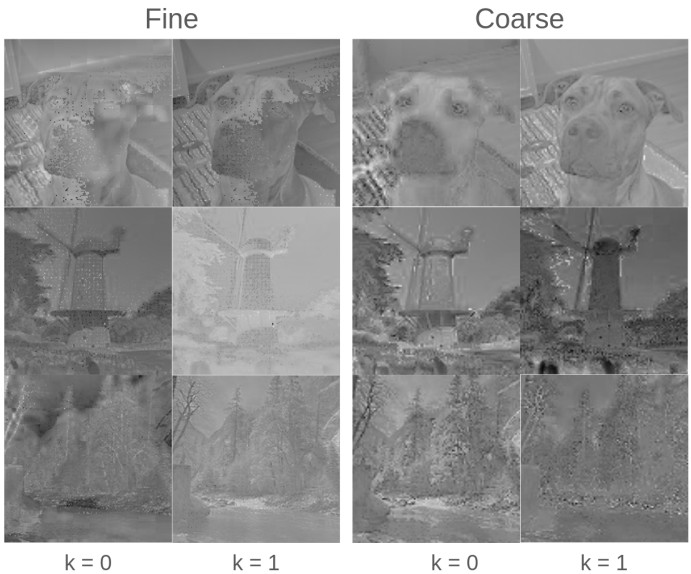

k = 0  k = 1  k = 0  k = 1

Figure 12: This figure shows the learned scalars at grid points in the MPE as a gray scale image. This allows us to get greater insight into what the encoding is doing. We plot results for both the fine grid encoding (left) and the coarse grid encoding (right). Each encoding has a single layer and two learnable parameters at each node. The $0^{t}h$ parameter is shown in the left column and the $1^{st}$ parameter is in the right column. We find that the encoding is learning representations similar to the regressed image. We also note that the fine grid encoding has small white patches in the middle most likely from where no pixel coordinate landed in the ground truth image.

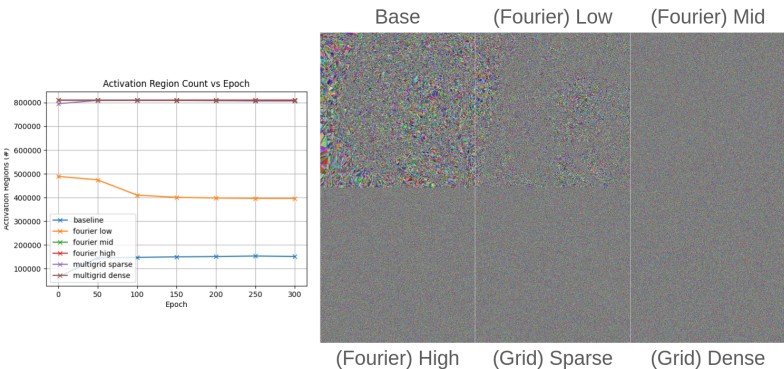

Figure 13: In this figure we plot the total number of activation regions found in the network across encodings as well as the activation region plotted per pixel of the regressed image. Note that the multigrid parameters are slightly different in this instance, with the sparse encoding having 8 layers with 2 parameters at each grid cell, while the dense encoding has 24 layers with 2 parameters at each grid cell. We find that even though the grid encoding seems to be learning the image (Figure 12, the baking network still has increased activation.

## D HYPERPARAMETERS

This section lists the hyperparameters used by encodings in each experiment. Table 2 shows the hyperparameters used in the ImageNet evaluation. These parameters were found using Optuna Akiba et al. (2019) to reflect real world training. The tuner was set to maximize the PSNR value between the network output and ground truth. Table 3 gives the hand selected parameters that were used to investigate how adjusting the parameters affects the output. We see that increasing the FFE's frequency improves the PSNR, but there are diminishing returns, as there is a smaller difference

| Encoding | Hyperparameters | | | | Results |
| --- | --- | --- | --- | --- | --- |
| | Learning Rate | Epochs | Batch Size | Grid Parameters | PSNR |
| Multigrid | 0.3932 | 100 | 10 | $k = 3, L = 2, x^{\pm} = [96, 277]$ | 45.28 |
| Fourier | 0.3865 | 100 | 92 | $L = 6$ | 33.47 |
| Baseline | 0.2394 | 100 | 10 | N/A | 29.94 |

Table 2: This table shows the results of hyperparameter tuning using the Optuna library. 30 trials were conducted where the tuner was able to select values for learning rate, batch size, and encoding specific parameters. The tuner sought to maximize the peak signal-to-noise ratio (PSNR) between the regressed and true image. The multigrid parametric encoding (MPE) is able to maintain a large lead in PSNR over the Fourier feature encoding (FFE) and baseline. These parameters were used in a larger sweep to characterize how the encodings behave across a wide class of images.

| Encoding | Hyperparameters | | | | Results |
| --- | --- | --- | --- | --- | --- |
| | Learning Rate | Epochs | Batch Size | Grid Parameters | PSNR |
| Low (FFE) | 100 | 300 | 32 | $L = 4$ | 25 |
| Mid (FFE) | 100 | 300 | 32 | $L = 8$ | 35 |
| High (FFE) | 100 | 300 | 32 | $L = 16$ | 36.5 |
| Coarse (MPE) | 100 | 300 | 32 | $k = 2, L = 1, x = 100$ | 34.5 |
| Fine (MPE) | 100 | 300 | 32 | $k = 2, L = 1, x = 200$ | 41.5 |
| Baseline | 100 | 300 | 32 | N/A | 20 |

Table 3: This table shows the parameters used in scaling experiments on the different encodings. We compare 3 FFEs of increasing frequency and 2 MPEs with fine or coarse grids (e.g., greater than or less than the number of pixels in the image). The PSNR column reports the regressed image against truth at the end of training. For details on training, please see *Experimental Setup*.

between Mid and High than there is between Low and Mid. The MPE, in contrast, works well for both the coarse and fine configurations. Lastly, Table 4 gives the Optuna found encoding parameters for the 3D implicit surface regression. The tuner sought to minimize the training loss. Results for the implicit surface evaluation can be found in Appendix E.

| Encoding | Hyperparameters | | | |
| --- | --- | --- | --- | --- |
| | Learning Rate | Epochs | Batch Size | Grid Parameters |
| Armadillo (Base) | 0.92224 | 4000 | 13187 | N/A |
| Armadillo (FFE) | 0.78930 | 4000 | 9799 | $L = 7$ |
| Armadillo (MPE) | 0.99469 | 4000 | 10903 | $k = 1, L = 1, x = 44$ |
| Buddha (Base) | 0.92224 | 4000 | 13187 | N/A |
| Buddha (FFE) | 0.78930 | 4000 | 9799 | $L = 7$ |
| Buddha (MPE) | 0.78445 | 4000 | 9267 | $k = 2, L = 2, x^{\pm} = [38, 102]$ |
| Dragon (Base) | 0.92224 | 4000 | 13187 | N/A |
| Dragon (FFE) | 0.78930 | 4000 | 9799 | $L = 7$ |
| Dragon (MPE) | 0.80905 | 4000 | 9989 | $k = 2, L = 2, x^{\pm} = [33, 136]$ |

Table 4: This table provides encoding parameters used in the evaluation of 3D implicit surfaces from meshes (see Appendix E). Parameters were found using Optuna minimizing the training loss. Grids were restricted to no more than 3 layers deep and 3 trainable parameters per node.

# E 3D IMPLICIT SURFACE

The NTK spectrum was evaluated on 3D implicit surface regression to demonstrate that the theory holds in 3D problems as well. The implicit surface was training using an 8 layer MLP with 256 neurons pre hidden layer. The final layer was passed through a sigmoid and then loss was evaluated by binary cross entropy. The input mesh was scaled to normalize its longest axis between 0 and 1. Points were randomly sampled each epoch in the scaled mesh's bounding box and input to the network. An output of 0 meant the point was outside the mesh, and a 1 meant it was inside. To visualize the surface, a ray marching method was used to render the 0.5 level set of the function,

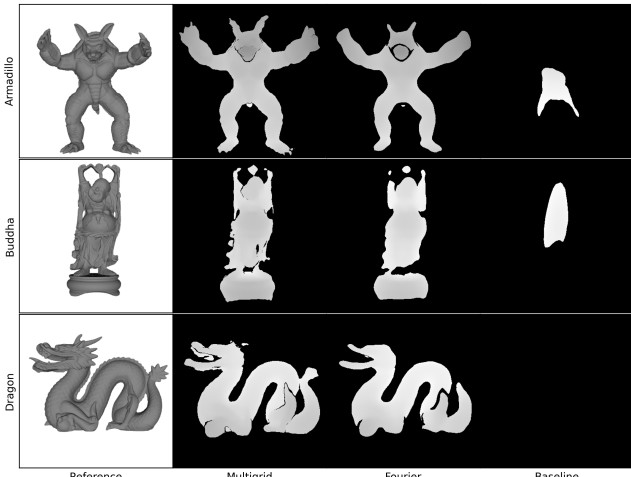

Figure 14: This figure plots the depth field at the 0.5 level set of the learned implicit surface. This rendering shows the network's ability to capture the structure of the input mesh. We find that the MPE outperforms the FFE and baseline in all scenarios, with baseline struggling to capture any recognizable features of the original mesh.

which is expected to represent the surface of the mesh. Again, there is no distinct separating between training and test data, and optimization was done using stochastic gradient descent Kiefer & Wolfowitz (1952).

As expected, we see that the MPE outperforms the FFE and baseline in terms of fine detail. The baseline network fails to even learn a recognizable structure in all cases. The FFE is able to represent some structure, but struggles with finer detail. The corresponding average eigenvalue spectrum was plotted in the main paper in Figure 5b. The hyperparameters used in each network can be found in Table 4.

