# OpenReview forum: "How Learnable Grids Recover Fine Detail in Low Dimensions: A Neural Tangent Kernel Analysis of Multigrid Parametric Encodings"
_ICLR.cc/2025/Conference — ICLR 2025 Poster_

### Official Review · Reviewer_L1id · 2024-11-01

**Soundness:** 3
**Presentation:** 2
**Contribution:** 2
**Rating:** 3
**Confidence:** 1

**Summary:**

The authors look at the Multigrid Parametric Encoding through the lens of Neural Tangent Kernel analysis.  They find that MPE have a raised eigenvalue spectrum compared to baseline encodings.

**Strengths:**

The approach seems straightforward and shows an improvement over the baseline.

I appreciate the theoretical contribution and subsequent evaluation of another approach.

**Weaknesses:**

I feel like the choice of the grid is not discussed enough.  There is either the choice of having a regularly spaced grid as well as having a learnable one.  There would at least be some intermediate option as well, like having an irregular grid, as it is done in the fast multipole method.  There it has also been observed that using a regular grid leads to unstable results.  Hence, it's not convincing that the grid actually needs to be learnable to be useful, like the authors claim.

The baseline network is not really defined.  After reading through the paper, I am still not sure what you mean by that and how you trained it.

While the analysis performed appears sound, it begs the question of what the take-away message is.  Does MPE always outperform FFE?  Figure 2 is missing the eigenvalue spectrum for FFE.

As it stands, the examples seem a bit cherry-picked.  It would improve the paper, if the authors could provide an average eigenvalue spectrum for both approaches considered that gives an idea of how it generalizes.

**Questions:**

If you use non-parametric networks for encoding an image, maybe it would be useful to also include comparisons to classical image compression algorithms and show their signal to noise ratio?

What does Identity correspond to in the plots?

It would be nice to see the actual learned grid superimposed on the image itself.

---

> ### Author Response · Authors · 2024-11-14
> **Response to Weaknesses and Questions**
>
> We appreciate the reviewers feedback and thank them for their time. We apologize for any confusion in the grid. The grid structure itself is not learnable and is instead set by the hyperparamters L and d (now h to avoid conflict with the input dimensions). At each intersection point of the grid there are learnable parameters that are trained using gradient descent and backpropagation. The structure of this grid is shown in figure 1 and examples of these learned parameters are shown in Appendix C. We consider both a uniform grid of decreasing cells as well as a logarithmic grid in our evaluations. The baseline is a similar point of confusion as it is just the same as the identity encoding. We address this in the question answer below.
>
> The takeaway from this analysis is that the MPE often outperforms the FFE and should be considered in more applications. As pointed out by reviewer pL4m, the MPE has been mainly used in the graphics community. Our work addresses the theoretical underpinning of the MPE in hopes that it sees a wider adoption in areas such as scientific computing. The MPE is often easier to use and does not suffer from the alias problems found in the FFE (Tancik et al., 2020). The MPEs main limitation is the memory requirement for the grid, but this has already been addressed by sparse (Hadadan et al., 2021) and hashgrid (Muller et al., 2022) representation that reduce memory requirements without changing our analysis. The MPE also is not suited for high dimensional problems as the scaling is poor; however, these domains often find the MPE unnecessary because the spectral bias problem is not present. This is why you will see the FFE used as the positional encoding for Transformers but not the MPE.
>
> We present evaluations on a random 100 image subset of ImageNet to show the generalization of our analysis on a wide variety of images. This is shown in Table 1 and Figure 5. The spectra are averaged across each evaluation and show that the MPE consistently outperforms the FFE and network without any encoding. We have also expanded our results with 3D implicit surface representations on meshes from the Stanford graphics dataset and show that the MPE still produces the highest eigenvalue spectrum overall. Muller et al., 2022 has already shown the broad applicability of the MPE. Our work provides the theoretical underpinning for why the encoding works so well.
>
> We appreciate the reviewer’s feedback in strengthening our paper. As we work on a new draft, are there any shortcomings that can be addressed to improve your view of our contributions? Our paper provides valuable insight into how the MPE functions, and it is our hope that this understanding will motivate future work in further optimizing the encoding as well as extend the MPE’s adoption beyond the graphics community.
>
> Questions:
> 1. Would it be useful for comparison against other image compression algorithms?
>
> We appreciate the reviewer's suggestion but believe that such a comparison would detract from the goal of the original paper. We are using the 2D image regression problem to illustrate the performance of the different results and provide provide empirical evaluations for the theory. We are not necessarily trying to find the most optimal image compression algorithm. We point the reviewer to Tancik et al., 2020 and Muller et al., 2022 for the broad applicability of these encodings.
>
> 2. What does identity correspond to in the plots?
>
> Identity and baseline both refer to a network without any encoding. The change in naming is confusing and we have remedied this in the current draft by referring to this network as the baseline throughout. We also made this explicit at the start of section 5.
>
> 3. It would be nice to see the actual learned grid superimposed on the image itself.
>
> Figure 1 shows the structure of the grid overlayed on an example image; however, this grid is much finer in practice. The structure of the grid itself is set with hyperparameters before training. Appendix C shows what the learned grid parameters look like at the end of the training. If there are additional figures that the reviewer would find helpful, we would be happy to add them in a future draft.

---

> > ### Author Response · Authors · 2024-11-17
> > **Updated Draft**
> >
> > We have uploaded a new draft that addresses the concerns and question of the reviewer. We thank the reviewer for their time and feedback, which has helped us improve the impact of our work. Important changes that we would like to highlight are:
> > - The removal of the identity label to avoid confusion. All figures and tables have been updated to use baseline for a network with no encoding. A description of the baseline is also given in "Experimental Setup" on line 207.
> > - We have expanded our results to include 3D implicit surface regression to show the broad applicability of our analysis. Figure 5(b) shows that the same trends in eigenvalue spectrum holds across encoding as the 2D image regression. Further details can be found in "OccupancyNet Analysis" at the bottom of page 9 and in Appendix E.
> >
> > We ask that the reviewer read through these change and reevaluate their impression of our work. This paper provides a novel application of the NTK to compare and analyze encodings across multiple problems. This insight is invaluable to undestanding how encodings affect network performance. We believe that this understanding will improve adoption of the MPE and lead to better performance in the future.
> >
> > Please review the updated draft and responses to the original questions. We would be happy to address any additional concerns or questions.

---

> > ### Comment · Reviewer_L1id · 2024-11-22
> >
> > I appreciate the authors response.  It appears that the paper relies heavily on knowledge about NTK and FFE, two topics that I am not familiar with.  I have thus changed my confidence score, to better reflect this.  Perhaps this would be better to submit to a computer graphics venue, as those topics are more prominent in those communities (though I don't necessarily think that it's a bad fit for ICLR).

---

> > > ### Author Response · Authors · 2024-11-24
> > >
> > > We thank the reviewer for their feedback and would be happy to provide additional resources on the NTK and FFE if the reviewer is interested. We appreciate the suggestion to submit to a graphics focused venue; however, we believe that this work is of great interest to the ICLR community. Our analysis provides a clear and easy means of comparing low dimensional MLPs, which are still integral to modern deep learning. The original analysis on FFEs using a NTK perspective was published in Neurips (Tancik, et al. 2020), and so was the original NTK paper (Jacot et al., 2018). We also hope that our work improves adoption of the MPE outside the graphics community, as the encoding's strong performance and easy tuning could have wide applicability. For example, the recent interest in implicit neural representations (INR) (Essakine et al., 2024) could greatly benefit from MPEs, but often only the FFE is considered. To further strengthen this connection, we have included a reference to an INR survey paper on line 38 in the introduction. We thank the reviewer for helping us to highlight this important connection, and if we have addressed the reviewer's concern, we kindly request that the reviewer reevaluate our work and raise the score.

---

> ### Author Response · Authors · 2024-11-20
>
> Hello. We wanted to see if you have been able to read through our responses as well as the updated draft. We would be happy to address any other concerns and would appreciate any additional feedback.

---

> ### Author Response · Authors · 2024-11-27
>
> We would like to point the reviewer to recent responses to reviewer EUYS as well as updates to Section 3 in the paper, which explain the connection between the NTK and the learned features in greater detail. The other concerns and questions in your original review have been carefully addressed, including providing expanded evaluations in a more difficult 3D implicit surface domain. Furthermore, we believe that the latest manuscript is much stronger and better aligned with the ICLR audience's expectations. We thank the reviewer for providing feedback allowing us to improve the impact of our work. If you agree that these points have been addressed and there are no further concerns or questions, we kindly ask that you consider raising our paper's rating.

---

> ### Author Response · Authors · 2024-12-02
>
> We hope this message finds you well. As the rebuttal deadline approaches, we kindly ask that you share any additional feedback or clarifications on your comments. Your input is invaluable in helping us address concerns effectively and enhance the strength and impact of our work.

---

### Official Review · Reviewer_pL4m · 2024-11-03

**Soundness:** 4
**Presentation:** 4
**Contribution:** 3
**Rating:** 8
**Confidence:** 2

**Summary:**

This paper studies how Multi-grid positional encodings affect the spectrum of the Neural Tangent Kernel of coordinates-based MLPs, and show that it forms a provably better encoding (compare to Fourier) to recover high frequency details.

**Strengths:**

- The paper doesn't make assumptions about what the reader knows or doesn't know and redefines terms and gives clear examples as well as clear expressions for the tools and objects mentioned.
- Isolating the effect of the embedding size and the grid is really important theoretically and practically. I wouldn't necessarily have thought to ask for this analysis, but seeing it in the paper is definitely a strength.
- The work is to the point with a clear theoretical contribution backed by solid experiments.

**Weaknesses:**

- nit: in notations, it would be nice to discuss why $d \leq 3$ is important for the theoretical analysis and the limitation.
- nit: in notations, $f_\theta(x)$ is not a function, it's an expression. Same for $\gamma(x)$. Btw, they shouldn't both use $x$ as an input.
- FFE NTK contribution: line 256 says "we derive the kernel for MPEs and FFEs", while line 134 says "Previous analysis of the NTK for coordinate based MLPs has been restricted to FFEs". I think it's really important to clarify what exactly the contribution is when it comes to the NTK of FFE based MLPs. If the latter is what is actually the case, then a reference to this work would be appropriate.
- Use of PSNR: because PSNR is an MSE-based metric, it is actually not really sensitive to fine details. I think using a metric like HFEN or a multiscale-SSIM would make more sense in this context.
- Large-scale realistic experiments: while I do think the current state of the experiments is really solid, for a more impactful work I think larger scale experiments with NeRFs or other SotA models would be beneficial.

**Questions:**

- in Fig. 1, is $d=2$ dimension of the input in the notations or 3 as suggested by the input of the MLP? Either way it's a bit confusing
- In theorem 1, the result states that the spectrum of NTK of MLP+MPE is uniformly lower bounded by the spectrum of NTK of MLP, could it be possible to quantify given some assumptions how much above it it is? For now we have an empirical answer.
- What's unclear to me after reading this paper is: since there is a consensus in computer graphics that MPE is better + it seems to work so well out-of-the-box in this work, why isn't everyone (typically in scientific ML) using MPE?

---

> ### Author Response · Authors · 2024-11-14
> **Response to Weaknesses and Questions**
>
> We greatly appreciate the time spent reviewing and providing feedback for our paper. We are working to address the notation issues in a new draft. To address the weaknesses, we have included a short description highlighting why the low dimensionality is important. The notation used for the kernel regressions and encodings was chosen to mirror previous works (Tancik et al., 2020; Wang et al., 2020), but we will change the x for kernel regression to be capitalized to represent the entire dataset (as defined in the notation section). Thank you for pointing out the lack of clarity in the original derivation description. The FFE's NTK was well established in Tancik et al., 2020, but the equations were given in a different form that makes it difficult to compare directly to our results. Our derivation does not change their original analysis, but it allows us to more easily compare the structure of the two kernels. We've updated the sentence at the start of section 4 to highlight this distinction. We also thank the reviewer for highlighting the limitations with PSNR. Our results follow the evaluations presented in Tancik et al, 2020, Hadadan et al., 2021, and Muller et al., 2022 which use PSNR. We will reevaluate the results using these new metrics and update a future draft. To address the generalization, we have completed additional evaluations on a 3D implicit surface problem using OccupancyNet (Mescheder et al., 2018) on meshes from the Stanford graphics dataset.
>
> Thank you for your time and feedback. While we update the draft, are there additional details that you would like to see included?
>
> Questions:
> 1. What is d in figure 1?
>
> The d in Figure 1 refers to the depth parameter of the grid, which describes the number of learnable parameters at each grid corner. As you point out, however, it seems that this name is overloaded. To address this, we have update the value to h in the current draft. Table 1, Table 2, and Figure 1 have been updated to reflect this. We thank the reviewer for helping to approach the notation used in the paper.
>
> 2. Is it possible to quantify the improvement of the MPE over the MLP with no encoding?
>
> Sadly, there is no readily available way to quantify the improvement of the MPE spectrum other than providing a lower bound. To address this, we included Figure 2. Though out of scope for the current work, this finding provides an efficient way for future work to optimize or design new parametric encodings as the kernel only needs to be evaluated on the learnable grid. The minimum eigenvalue could then be used during a hyperparameter search or to compare different grid interpolation schemes. Though we are not able to find an exact quantity for this gap, the method laid out is widely applicable to evaluation of parametric encodings.
>
> 3. Why isn't everyone using MPE?
>
> This is a good question and was a major motivation for our work. MPEs have strong performance and do not suffer from the aliasing problem found with FFEs (Tancik et al., 2020). One limitation is the amount of memory needed to store the grid, but this problem has been largely addressed by a sparse grid representation (Hadadan et al., 2021) and a hashgrid representation (Muller et al., 2022). It's important to note that these only affect the storage and do not change the analysis presented in this paper. The MPE has been largely published in the graphics community, but we hope to see more papers using MPE in other communities. We also note that there has been some use of the MPE in scientific machine learning applications (Huang and Alkhalifah, 2023), and we hope that our work will improve adoption by providing a more theoretical basis for the MPE, which was already explored for the FFE.

---

> > ### Comment · Reviewer_pL4m · 2024-11-14
> >
> > Thanks for engaging in the review process respectfully and benevolently.
> >
> > I am happy with the response provided, and will keep my score.
> >
> > - Could you please include in writing here the content of your modifications?
> > - Re the memory limitation of MPE, I think it would be nice to mention it in the paper

---

> > > ### Author Response · Authors · 2024-11-17
> > > **Updated Draft**
> > >
> > > We have uploaded a new draft which incorporates the reviewer's feedback. We thank the reviewer for their helpful comments to improve the impact of our work. Important updates the we would like to highlight are:
> > > - Figure 1 has been updated to not overload the meaning of $d$. $k$ is now used to align with Equation 3. Line 208 has been updated to consider $k$ as the number of learnable scalars at each grid intersection instead of the index. All tables (Appendix D) should be in alignment with this definition.
> > > - Evaluations using multi-scale SSIM have been included and can be found in Table 1. The previous hyperparameter tables were moved to Appendix D with their new location highlighted in the "Experimental Setup" section. We find that the new metric follows the same trend as PSNR, and thank the reviewer for improving the evaluations in our work.
> > > - The paper now includes evaluations on 3D implicit surface regression. Qualitative results and problem setup are found in Appendix E due to space. A brief analysis can be found at the bottom of page 9 in a new "OccupancyNet Analysis" section. The average eigenvalue spectrum of the NTK for this evaluation can be found in Figure 5 (b).
> > > - We have highlighted the trade-off in terms of memory in the new draft. It is mentioned in the introduction on line 71 and in the background section n line 210-212.
> > >
> > > We again thank the reviewer for their time and are happy to address any other concerns or questions that they may have.

---

### Official Review · Reviewer_uZqD · 2024-11-04

**Soundness:** 2
**Presentation:** 2
**Contribution:** 2
**Rating:** 5
**Confidence:** 2

**Summary:**

The paper provides a theoretical and empirical analysis demonstrating that Multigrid Parametric Encodings (MPEs) improve neural network performance in learning fine details and handling discontinuities. By deriving the neural tangent kernel (NTK) for MPEs, the authors show that multigrid encoding elevates the NTK's eigenvalue spectrum compared to coordinate-based MLPs, explaining why MPEs capture detail more effectively. Their analysis isolates this improved performance to MPEs’ learnable grid structure rather than the embedding space alone, a distinction from Fourier Feature Encodings (FFEs). Empirical results on 2D image regression with ImageNet data reveal that MPEs, especially those with smaller grid cells, achieve higher Peak Signal-to-Noise Ratio (PSNR) scores, indicating better detail preservation than both FFE and baseline MLP approaches.

**Strengths:**

1. The paper has a well-established theory.
2. The paper is well structured.
3. The authors aim to fill the gap in understanding why MPEs improve a network’s performance through the structure of their grid and not their learnable embedding.

**Weaknesses:**

1. The paper offers limited novelty and applicability. The authors demonstrate that MPEs enhance network performance through grid structure rather than learnable embeddings by analyzing a lower bound on the eigenvalues of the Neural Tangent Kernel (NTK). However, they use well-known methods to address the problem, which limits the work's novelty.
2. Most of the paper is focused on describing the existing theory which is well described in literature.
The evaluation process is insufficient; in my opinion, images from a single dataset (ImageNet) are not enough to thoroughly validate the theory.
3. In Equation (9), derivatives are denoted by a single prime symbol without specifying the parameter with respect to which they were calculated, creating confusion and potential errors in further derivations, which makes the paper difficult to follow.
4. Furthermore, the analysis focuses primarily on a single-layer MLP network. While the authors state that the theory can be easily extended to deeper networks, they do not provide this extension, even in the appendix. The structure and width of individual layers in finite-width networks introduce variability in the NTK. For example, deeper layers in finite networks may capture complex feature hierarchies, while shallow layers contribute differently to the NTK. This dependency adds analytical challenges, as each layer’s contribution can influence the network’s performance and generalization in unique ways. This is the main reason why the evaluation process should also be conducted on deeper neural networks.

**Questions:**

1. Can the presented theory be applied to other neural network architectures, such as Convolutional Neural Networks (CNNs)?
2. Do the presented figures show results only for single-layer fully connected neural networks?

---

> ### Author Response · Authors · 2024-11-14
> **Response to Weaknesses and Questions**
>
> We thank the reviewer for their time and the important feedback. We address your questions below, but would first like to respond to the weaknesses presented. We agree that the neural tangent kernel is a well established method; however, we see this as a strength of our work rather than a detriment to it. Previous work (Jacot et al., 2020; Tancik et al., 2020; Wang et al., 2020; Yang and Salman, 2019) has established that the NTK is representative of real networks in use. The novelty of our work is in the application of the NTK to the multigrid parametric encoding. This analysis provides invaluable insight into the function of the encoding, allowing for better fine-tuning and broader application in the machine learning community. The establishment of the NTK in the literature also lends itself to the generalization of our results on ImageNet. Our evaluation on a wide variety of images shows that the spectrum holds. Previous work by Tancik et al., 2020, Muller et al., 2022, and Hadadan et al. 2021 have shown that the encodings work well on a wide variety of applications, but the eigenvalue spectrum for the MPE was never established. To strengthen our results, we have evaluated the NTK on OccupanyNet (Mescheder et al., 2018) to show that the spectrum holds for implicit surface representations in 3D. Meshes from the Stanford graphics dataset, as used by Tancik et al., 2020, show that the MPE produces a larger eigenvalue spectrum in the 3D case as well. We are preparing these plots for an upcoming draft. Reviewer pL4m corroborates this stance that the ImageNet evaluations are sufficient. The draft also updates the notation in equations 9, 12, and 14 to make the derivative more explicit as they are with respect to the model parameters.
>
> We again thank the reviewer for their time. Are there any other aspects of the paper that we can address to strengthen your impression of the work? We believe that our analysis is important for the wider adoption of the MPE and better use of encodings in machine learning applications.
>
> Questions:
> 1. Can the theory be applied to other architectures like CNNs?
>
> Yes, the theory can be applied to most common architectures as shown by Yang and Salman, 2019. Our analysis however is purposely restricted to MLPs as described at the start of the introduction. Often there are applications in graphics or scientific machine learning where the network needs to be both small and fast to evaluate. There is also typically not regular structure that can be applied to CNNs, so MLPs are used.
>
> 2. Do the presented figures show results only for single-layer, fully connected networks?
>
> We apologize for the confusion. The derivation in section 4 focuses on a single layer as extension to multiple layers is trivial, but the number of terms quickly become unwieldy, adding little benefit to the reader’s understanding of our work. As explained in “Experimental Setup”, the networks used contain two hidden layers with a width of 512 each. Evaluations on these networks are use for the plots in all figures. Our additional evaluations on 3D implicit surface regression using a network with 6 fully connected hidden layers with a width of 256 each. In this case, we find that the same eigenvalue spectrum trends hold, with the MPE greater than the FFE which is, in tern, greater than the baseline, unencoded network.

---

> > ### Author Response · Authors · 2024-11-17
> > **Updated Draft**
> >
> > We have uploaded a new draft that addresses the questions and concerns of the reviewer. We thank the reviewer for their time in helping to improve the impact of our work. The new draft remedies the original concern with notation and expands the empirical results to 3D implicit surfaces, which used deeper (8 layer) networks. Key points to highlight are:
> > - Equations 9, 12, and 14 have been updated to show that the derivative is with respect to the model and grid parameters $\theta$.
> > - A new "OccupancyNet Analysis" section has been added to results at the bottom of page 9. This section shows that the same trends found in the 2D image regression problem hold for 3D implicit surface regression. Figure 5 includes the average eigenvalue spectrum found. These new results show that the theory and empirical results are in strong alignment, and that the NTK analysis hold across multiple domains. The network uses was also much deeper with 8 layers, which addresses the original concern of limited evaluations.
> > - Appendix E has been included to show the results from the 3D implicit surface regression
> >
> > Given our responses and updated draft, we ask that the reviewer reevaluate their impressions of our work. We have expanded on previous works in the literature by computing the eigenvalue spectrum for both encodings. Previous work only showed the spectrum in the 1D or 2D domain and only for the FFE. These evaluations and understandings also were not directly transferrable to the MPE, as its improvement draw from the learnable grid. Our contributions provide a deeper understanding of how to compare encodings and analyze their performance.
> >
> > Are they are any other concerns or questions that we can address?

---

> ### Author Response · Authors · 2024-11-20
>
> Hello. We wanted to see if you have been able to read through our responses as well as the updated draft. We would be happy to address any other concerns and would appreciate any additional feedback.

---

> ### Author Response · Authors · 2024-11-24
>
> Hello. Are there any other concerns or questions that we can address? We believe that the updated draft has improved the presentation and demonstrated that the theory holds well for deeper networks (both 2 layer and 8 layer). If possible, it would be helpful to receive any additional feedback soon, so that there is enough time to respond before the deadline on November 26th. Thank you again for the time and effort taken to review our work.

---

> > ### Comment · Reviewer_uZqD · 2024-11-26
> > **Comment by Reviewer**
> >
> > Thanks for the clarification. I raise the score.

---

> > > ### Author Response · Authors · 2024-11-27
> > >
> > > We thank the reviewer for their response and going through the updated draft. Are there any other concerns or questions that we can answer? The inclusion of additional evaluations on occupancy net shows that the theory holds across domains and network depths. Our work provides an important explanation for why the MPE consistently outperforms the FFE, while also demonstrating an easy means of comparison through the empirical evaluation of the NTK. By establishing this connection in the literature, we hope to increase adoption of the MPE, especially as the ICLR and broader machine learning community continues to use low dimensional MLPs in many graphics, machine learning, and implicit neural representations applications.

---

> ### Author Response · Authors · 2024-11-27
>
> Thank you for your thoughtful feedback on our submission. We greatly appreciate the time and effort taken to review our work. We have carefully addressed the concerns and questions in your original review, and believe that the latest manuscript is much stronger and better aligned with the ICLR audience's expectations. If you agree that these points have been addressed and there are no further concerns or questions, we kindly ask that you consider raising our paper's rating.

---

> ### Author Response · Authors · 2024-12-02
>
> We hope this message finds you well. As the rebuttal deadline approaches, we kindly ask that you share any additional feedback or clarifications on your comments. Your input is invaluable in helping us address concerns effectively and enhance the strength and impact of our work.

---

### Official Review · Reviewer_EUYS · 2024-11-04

**Soundness:** 2
**Presentation:** 1
**Contribution:** 2
**Rating:** 5
**Confidence:** 4

**Summary:**

The article proposes a multi-grid parametric encoding approach
to capture high-frequency information in images.
Through a neural tangent kernel analysis angle,
it is proven that the proposed encoding can learn an NTK
kernel with a higher eigenvalue spectrum than using no encoding.
The superior performance of this approach is further analyzed
in detail by separating the contribution from the learnable grid, and
the embedding space. On the image regression task of ImageNet,
the proposed encoding ourperforms Fourier feature encodings.

**Strengths:**

- It is an interesting idea to analyze the impact of different
encoding schemes of a coordinate of an image using the spectrum
in the  NTK framework to make the theory concise.
- The proposed multi-grid parametric encoding works very well in
the image regression task.

**Weaknesses:**

- The writing of the article should still be improved
for the reviewer to understand the methodology. Please clarify the questions below.
- Conceptually, it is still unclear how the spectrum of the NTK
is related to the high-frequency information of an image, i.e.
the large modes of the Fourier transform of a signal X = (x_1,...,x_N) of length N.

**Questions:**

- It is unclear in the definition (1), what is the dimension of x,  is it in R^d?
If so what about the dimension of the output \gamma_F(x), is it 2d x L ?
- It is not clear what is the notation round + in eq. 3 means.
- eq. 7 is hard to understand. If K_NTK is a matrix of size NxN, would the size
of Q be also NxN? In this case, why it makes sense to write Q ( f_theta(x,t) - y ) if y is not on dimension N?
- Is there an expectation missing in eq. 9 to define the K_NTK, as in eq. 4?
- The statement of Theorem 1 is not so clear, it would be better to write directly what lambda_i^MPE means as in the proof.
What is the size of the training set X^n in this proof? Isn't it the X in the notation section?

---

> ### Author Response · Authors · 2024-11-14
> **Response to Weaknesses and Questions**
>
> Thank you for your review and providing feedback to improve the impact of our work. We apologize for the confusion in our presentation and seek to address your questions below and in a new draft. We would, however, like to highlight comments from reviewers uZqD and pL4m, which emphasize the clarity and structure of our paper. In response to the weaknesses presented, the connection between the eigenvalues and high frequency information has been explored in multiple works (Tancik et al., 2020; Wang et al., 2020) and is explained in section 3. Equation 7 shows that the error decays along each eigenvector based on the magnitude of the corresponding eigenvalue. If the eigenvalue is larger, then the decay is faster. The smaller eigenvalues inherently match to higher frequency information, as explained in Basri et al., 2019. Beyond, the answers below, is there any additional information to be included in the paper to improve the reviewer's understanding of the background?
>
> While we fix these notational issues, are there other concerns that would improve your confidence in our work? Having a stronger theoretical understanding of encodings is important to the machine learning community, as it allows for the creation of improved coordinate based MLPs and wider adoption of the MPE. Our work leveraging the NTK provides such an important analysis.
>
> Questions:
> 1. Dimensions in Equation 1.
>
> Thank you for catching this omission. The dimension of x is as you have stated as it in the original input to the network. The output dimension is then $2dL$. We have included these values next to equation 1 for clarity.
>
> 2. Notation in Equation 3.
>
> The plus symbol is a common notation for concatenation.
>
> 3. Equation 7 is hard to understand. What are the dimensions of y?
>
> We apologize for the confusion. y in this context all of the training targets concatenated, so it is not just the output dimension. It is the output dimension multiplied by the number of samples. We've included the dimension explicitly in the "Spectral Bias" section and use a capital letter to denote that it’s the entire dataset.
>
> 5. Is there an expectation missing in equation 9?
>
> We thank the reviewer for pointing this out. We have updated equation 9, 12, and 14 to include the expectation.
>
> 6. Theorem 1 is unclear.
>
> We have removed the $n$ superscript to align more with the notation description. You are correct in that the two are the same. Lambda is a common notation for the eigenvalue of a matrix, and $\lambda_i^{MPE}$ is simply the ith largest eigenvalue of the NTK for the MPE contributions to the kernel shown in equation 13. We will highlight this connection in a new draft.

---

> > ### Author Response · Authors · 2024-11-17
> > **Updated Draft**
> >
> > We have uploaded a new draft that we hope addresses the concerns of the reviewer. We thank the reviewer for pointing out the confusing notation and have updated sections 3 and 4 to improve the presentation of our paper. Important changes are:
> > - The inclusion of encoding dimensions on line 182
> > - Updated notation in the NTK section of Related Works. We capitalize Y to align with the original description in the Notation section.
> > - Equations 9, 12, and 14 have been updated to include the expectation to align with equation 4.
> > - Theorem 1 has been updated to make it clear what matrix each eigenvalue corresponds to.
> >
> > We ask that the reviewer reevaluate their impressions of our work given these important updates. Beyond the addressed concerns, is there any other information that we can provide to improve your assessment?

---

> > > ### Comment · Reviewer_EUYS · 2024-11-26
> > > **some questions**
> > >
> > > Thank you very much for your answers. I appreciate your modifications of the main text to make it clearer.
> > >
> > > I still have some questions (based on your revised version):
> > > - in eq. 12, your computation of the kernel NTK MPE, does not seem to be correct: do you need add all
> > > the contributions of the gradients from each cell to the NTK? As $\tilde{g}'$ is encoding the gradient of one of the cells in
> > > in eq. 3 , I guess you need to include different $\tilde{g}'$ at each location (0,0) ... (L,k). It seems that you have done this in eq. 13, but again the notation is very confusing.
> > > - I still do not follow your answer: The smaller eigenvalues inherently match to higher frequency information, as explained in Basri et al., 2019. Actually, what do you mean by high-frequency information? In 2d image case, I think this is related to the high-frequency modes in 2d Fourier transform of an image, but it seems to me that Basri et al., 2019 is talking about something different.
> > > - You mentioned in the beginning of your section 5 that computing NTK for higher dimensional problems is prohibitive. Are you in this situation or not? How do you compute the NTK spectrum in your results in Fig. 4 and 5 if it were prohibitive ...
> > >
> > > I think that the writing should be improved to make the main text clearer to help your reviewer better understand your results (as well as its soundness). I would keep my current score unless other reviewers find the contributions significant and sound.

---

> > > > ### Author Response · Authors · 2024-11-27
> > > > **Question Responses and Paper Updates**
> > > >
> > > > We thank the reviewer for their response and for helping us to improve the understanding of our work.
> > > >
> > > > 1. Notation of equation 12 is confusing. Shouldn't it include terms for each grid cell?
> > > >
> > > > We apologize for the confusing description in our paper and thank the reviewer for helping to improve the presentation. You are correct. Equation 12 currently only considers a single grid cell, as stated on lines 300 and 301 of the current draft. The paragraph between Equation 12 and Equation 13 then explains the extension to multiple grids and learnable parameters. We hesitate to add more terms to Equation 12, as it is already 5 lines but agree that this connection is not obvious. We have updated the paragraph before Equation 12 to include the following:
> > > > "The gradients of the grid parameters are independent due to the concatenation, so we first compute the contribution of a single grid parameter. The full kernel is then the sum of each grid cell contribution."
> > > >
> > > > 2. What do you mean by high frequency information with respect to the eigenvalues and eigenvectors of the NTK?
> > > >
> > > > We thank the reviewer for pointing out this confusion in the paper. When we refer to high frequency information, we are referring to the finer details in the image and not necessarily a Fourier basis. This nomenclature mirrors what was previously used in Tancik et al., 2020 who analyzed the FFE using the NTK. The NTK represents the relationship between the gradients produced by each training data point. The eigenvalues of the NTK are then a basis for the relationship between the gradients. The "lower frequency" eigenvectors correspond to smoother relationships in the data, while "higher frequency" eigenvectors correspond to the finer relationships. Equation 7 shows that the error is reduced along the largest eigenvalues first. Basri et al., 2019 showed that in an MLP the eigenvalues decay in magnitude with higher frequency eigenvectors. This is what we see in practice with our results. The network is quickly able to fit the "smooth" features of the data, but struggles to fit the finer details without the help of the encodings. Conversely, we see that encodings which produce the highest eigenvalue spectrum produce the highest quality results. Our results show that the NTK's eigenvalue spectrum is a useful, quantitative way to compare the performance of the encoding on the dataset.
> > > >
> > > > We hope that this explanation helps the reviewer's understanding of the important background that we have built off of. To improve presentation, we have updated the final paragraph of Section 3 to read:
> > > >
> > > > "From Equation 7 we see that the loss has a dependence on the eigenvalues and
> > > > eigenvectors of the NTK and that larger eigenvalues will decrease the training
> > > > error faster along that dimension. Basri et al. (2019) found that eigenvalues
> > > > corresponding to higher frequency relationships in the dataset are lower for
> > > > coordinate-based MLPs. As a result, networks will fit lower frequency, smoother, relationships
> > > > much faster than they will the higher frequencies. By comparing the eigenvalue
> > > > spectrum produced by the composition of encodings and the NTK, we can evaluate
> > > > an encodings’ bias towards higher frequency features, which corresponds to an
> > > > increased number of large eigenvalues."
> > > >
> > > > We would also like to highlight feedback from other reviewers who have stated that our background section is clear and helpful (reviewer pL4m) as well as that the NTK is well established in the literature (reviewer uZqD). We hope that this change improves the presentation of our paper; however, if the reviewer finds that the section is still hard to understand given the background, we would be happy to make additional updates before the deadline.
> > > >
> > > > 3. You mention that computing the NTK of higher dimensional problems is prohibitive? With the inclusion of new results, is this still the case?
> > > >
> > > > We apologize for not making this update clearer. Since the initial submission, we worked to improve the computation of the NTK and added batch computing of matrix blocks to ease the memory requirements of the computation. The overall computation speed is still limited by the matrix decomposition, but evaluation on more data points has become easier. This comment was removed in the original revision to reflect this and avoid any confusion.
> > > >
> > > > We hope that these responses and updates help the presentation of our work and better represent the soundness of our contributions. If the reviewer has any other concerns or questions, we would be happy to address them.

---

> ### Author Response · Authors · 2024-11-20
>
> Hello. We wanted to see if you have been able to read through our responses as well as the updated draft. We would be happy to address any other concerns and would appreciate any additional feedback.

---

> ### Author Response · Authors · 2024-11-24
>
> Hello. Are there any other concerns or questions that we can address? We hope that the updated draft has addressed many of the initial concerns about notation and presentation. If possible, it would be helpful to receive any additional feedback soon, so that there is enough time to respond before the deadline on November 26th. Thank you again for the time and effort taken to review our work.

---

> ### Author Response · Authors · 2024-12-02
>
> We hope this message finds you well. As the rebuttal deadline approaches, we kindly ask that you share any additional feedback or clarifications on your comments. Your input is invaluable in helping us address concerns effectively and enhance the strength and impact of our work.

---

### Meta-Review · Area_Chair_vot2 · 2024-12-22

**Metareview:**

This paper compares NTK spectra of two implicit nets, one with Fourier feature encodings, the other multigrid, and find that the latter has more favorable spectral properties to recover high-freq details. While the reviewer opinions are mixed and the critical remarks meaningful, I think that the finding is important, and that the numerics are nice. It is valuable to have some sort of mechanistic understand for why one network is better than the other. What is missing is some stronger theoretical support. Theorem 1 is rather weak, and one would like have a quantitative statement which shows how the data distribution influences the spectral differences. I will recommend acceptance, but I would not be upset if the paper gets rejected. (+ Please see below about the interpretation of "learnable grids".)

**Additional Comments On Reviewer Discussion:**

EUYS asked about connections between the NTK spectrum and high-frequency information and made a number of useful remarks about clarity and writing. The authors responded with questions and clarifications but the reviewer identified still more confusing parts and math. There were also some questions about the computational complexity of experiments. uZqD had concerns about novelty and then some minor remarks about notation and architectures. After the authors responded the reviewer raised their score to just below the acceptance bar. pL4m championed the paper and pointed out clarity, self-containedness, importance of isolating effects of embedding size and the grid, ... , they mostly had minor remarks about notation, metrics, and scale. L1id asked about why the grid has to be "learnable", drawing on analogies with FMMs. L1id stated very low confidence and following his responses in the discussion I decided not to consider their score. Nonetheless, the question about "learnable" is good and I think the title is a bit misleading as it suggests something like "adaptive grids" which is a huge topic in numerical analysis. My recommendation based on carefully reading the discussion, considering reviewer confidence and expertise, and my own familiarity with the topic.

---

### Decision · Program_Chairs · 2025-01-22

Accept (Poster)